# Biocatalytic routes to stereo-divergent iridoids

Néstor J. Hernández Lozada [1], Benke Hong[1], Joshua C. Wood[2], Lorenzo Caputi[1], Jérôme Basquin[3], Ling Chuang[1], Maritta Kunert[1], Carlos E. Rodríguez López[1], Chloe Langley [1], Dongyan Zhao [2], C. Robin Buell[2], Benjamin R. Lichman [4] & Sarah E. O'Connor [1] ✉

Thousands of natural products are derived from the fused cyclopentane-pyran molecular scaffold nepetalactol. These natural products are used in an enormous range of applications that span the agricultural and medical industries. For example, nepetalactone, the oxidized derivative of nepetalactol, is known for its cat attractant properties as well as potential as an insect repellent. Most of these naturally occurring nepetalactol-derived compounds arise from only two out of the eight possible stereoisomers, 7S-cis-trans and 7R-cis-cis nepetalactols. Here we use a combination of naturally occurring and engineered enzymes to produce seven of the eight possible nepetalactol or nepetalactone stereoisomers. These enzymes open the possibilities for biocatalytic production of a broader range of iridoids, providing a versatile system for the diversification of this important natural product scaffold.

Iridoids are a class of monoterpene natural products that are produced in a diverse range of flowering plant species. Over one thousand distinct iridoid structures have been identified, and collectively, these compounds play wide-ranging roles in plant defense[1–3]. Moreover, many iridoids have pharmaceutical applications[4–8], including iridoid-derived monoterpene indole alkaloids such as the well-known anti-cancer drug vincristine. A key intermediate in the generation of most iridoids is nepetalactol **3**, which, along with its oxidized product, nepetalactone **4**, is well-known for its behavioral effect on cats[9,10]. In addition, the nepetalactone isomers are known to have stereo-divergent effects in repelling insects[11–14]. It is therefore desirable to unlock this structural diversity in order to apply it to this and many other potential applications downstream of **3**.

As with the vast majority of monoterpene pathways, iridoid biosynthesis begins with geranyl pyrophosphate (GPP), though in iridoid biosynthesis, GPP is invariably converted to geraniol, and then oxidized to 8-oxogeranial **1**[15–18]. **1** is then reduced by the enzyme iridoid synthase (ISY), which generates a reactive intermediate 8-oxocitronellyl enol **2** that is primed to cyclize to form the canonical

nepetalactol **3** scaffold, the precursor to all iridoids in plants[19]. **3** is further derivatized through oxidation, glycosylation, acylation and other enzymatic transformations to form the wide variety of iridoids and iridoid-derived compounds found in nature (Fig. 1A).

The nepetalactol **3** scaffold has three chiral centers, though only a subset of the 8 possible stereoisomers have been identified in nature. The stereochemistry of C7 is set first during the 1,4 reduction of 8-oxogeranial catalyzed by ISY; both 7S and 7R specific ISY enzymes have been identified[19,20]. The stereochemistry of the bridgehead carbons (C4a and C7a) are set during the subsequent cyclization of 8-oxocitronellyl enol **2**, the ISY product, to form **3**. The most common nepetalactol stereoisomers found in nature are 7S-cis-trans (7 S, 4aS, 7aR) **3a** and 7R-cis-cis (7 R, 4aS, 7aR) **3b′** (Fig. 1B)[21–23]. However, plants in the genus *Nepeta* (Fig. 1C) produce iridoids with varied stereochemistry at the bridgehead carbons: in addition to **3a**, iridoids with 7S-cis-cis (7 S, 4aR, 7aS) **3b** and 7S-trans-cis (7 S, 4aS, 7aS) **3c** stereochemistry are observed. We recently discovered a family of enzymes in *Nepeta mussinii* and *Nepeta cataria*, nepetalactol-related short chain reductases or NEPS, that, in tandem with a 7S-specific ISY, control the

[1]Max-Planck Institute for Chemical Ecology, Department of Natural Product Biosynthesis, Hans-Knoll Strasse 8, 07745 Jena, Germany. [2]Center for Applied Genetic Technologies, University of Georgia, Athens, GA, USA. [3]Max-Planck Institute for Biochemistry, Department of Structural Cell Biology, Am Klopferspitz 18, 82152 Martinsried, Germany. [4]University of York, Department of Biology, Centre for Agricultural Products, Wentworth Way, York YO10 5DD, UK. ✉e-mail: oconnor@ice.mpg.de

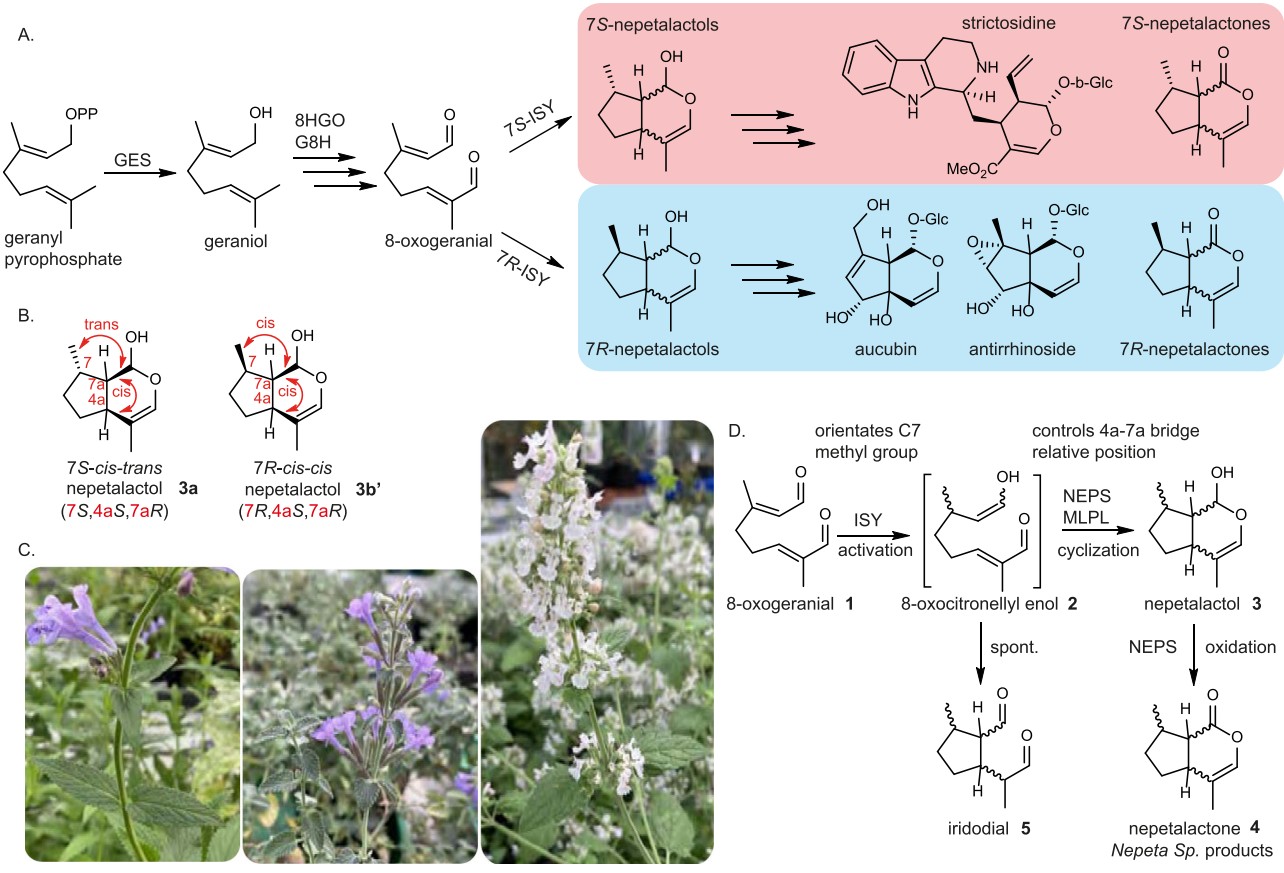

**Fig. 1 | Overview of the nepetalactol scaffold and its stereo-control in the *Nepeta* species. A** The early iridoid pathway takes two main directions based on the preferred stereochemistry of iridoid synthase (ISY). *7S*-ISY leads to biosynthesis products such as (7 *S*) nepetalactones found in most *Nepeta* species, as well as well-known indole alkaloid precursor strictosidine in *C. roseus* and other species (red). *7R*-ISY exists in other members of the Asterids clade, particularly the Lamiales order, where iridoid glucosides with this C7 stereochemistry is common (light blue). **B** Nepetalactol **3** stereochemistry can adopt 8 stereo-chemical configurations based on the relative positions of the C7 methyl group and the 7a-4a bridge. Most iridoids known in plants are based on the *7S-cis-trans* **3a** and *7R-cis-cis* **3b'** configurations (7 *S*, 4a*S*, 7a*R* and 7 *R*, 4a*S*, 7a*R*, respectively). **C** Three Nepeta species studied here, *N. sibirica*, *N. mussinii*, and *N. cataria*. **D** As for nepetalactol **3**, nepetalactone **4** also can exist in 8 stereo-chemical configurations. NEPS typically control the stereochemistry of the bridgehead carbons; in one instance a Major Latex Like Protein[24] catalyzes the formation of *7S-cis-trans* **3a** (7 *S*, 4a*S*, 7a*R*). NEPS also must be capable of cyclization and oxidation of both the 7 *S* and 7 *R* products of ISY in a selective manner to generate the full complement of stereoisomers.

stereo-selective cyclization of **2** to generate these three nepetalactol isomers[24,25] (Fig. 1D). In addition, the crystal structure of *N. mussinii* NmNEPS3, which catalyzes cyclization and partial oxidation to the *7S-cis-cis* nepetalactone **4b** isomer, was previously reported (PDB 6F9Q)[24]. NEPS belong to the NAD+-dependent short chain dehydrogenase family (SDR)[26] and share 65-95% amino acid identity among them as well as 62-73% identity to *Mentha x piperita* (-)-isopiperitenol/(-)-carveol dehydrogenase[27]. Some NEPS, although containing an NAD+ binding site, are redox inactive and catalyze only cyclization, others catalyze only the oxidation from nepetalactol to nepetalactone, while others are dual-function enzymes, catalyzing both stereo-selective cyclization and oxidation of various nepetalactols to nepetalactones. Enzymatic formation of diverse iridoid stereoisomers could lead to the synthesis of natural and derivatives of bioactive compounds in bio-catalytic and synthetic biology systems. The NEPS provide a starting point for developing stereo-selective cyclases to enable this.

Here, we set out to understand and expand the selectivity of NEPS by both gene discovery in *Nepeta sibirica* and enzyme engineering of NEPS from various *Nepeta* species: *N. sibirica*, *N. cataria* and *N. mussinii*. To compile a family of NEPS with the greatest possible natural catalytic variety, we identify six additional NEPS homologs in *Nepeta sibirica* (NsNEPSL, NsNEPS1A/B, NsNEPS2, and NsNEPS4A/B). *N. sibirica* was selected since it is the most evolutionarily distant member of the

*Nepeta* genus relative to *N. cataria* and *N. mussinii* that is readily available and native to the European Union[28]. Since we aim to rationally engineer cyclization and oxidation in NEPS as well as to better understand the mechanistic basis of catalysis, we additionally solve the structure of *N. sibirica* NsNEPS2 (PDB 7QUJ), which catalyzes the cyclization and oxidation to *7S-cis-trans* nepetalactone 4a (Supplementary Table 1). Furthermore, we expand the access to other stereoisomers via combinatorial biosynthesis, ultimately gaining access to seven of eight possible iridoid isomers and setting the stage for further derivatization into stereo-diverse iridoids. These enzymes can now be used in combination with existing microbial host systems to provide a set of divergent starting points for seco-iridoid and iridoid glucoside pathways[29-33].

## Results and discussion

### Cyclization and oxidative selectivity in NEPS

With a total of 19 wild-type NEPS sequences ranging from 64-96% sequence identity and two 3D NEPS structures to compare and contrast (Supplementary Figs. 1–2) we sought to understand the residues involved in the various catalytic activities observed. We first set out to decouple oxidation and cyclization activity of NEPS through muta-genesis to maximize biocatalytic flexibility. In addition to cyclization of 8-oxocitronellyl enol **2** to nepetalactol **3**, many NEPS catalyze the

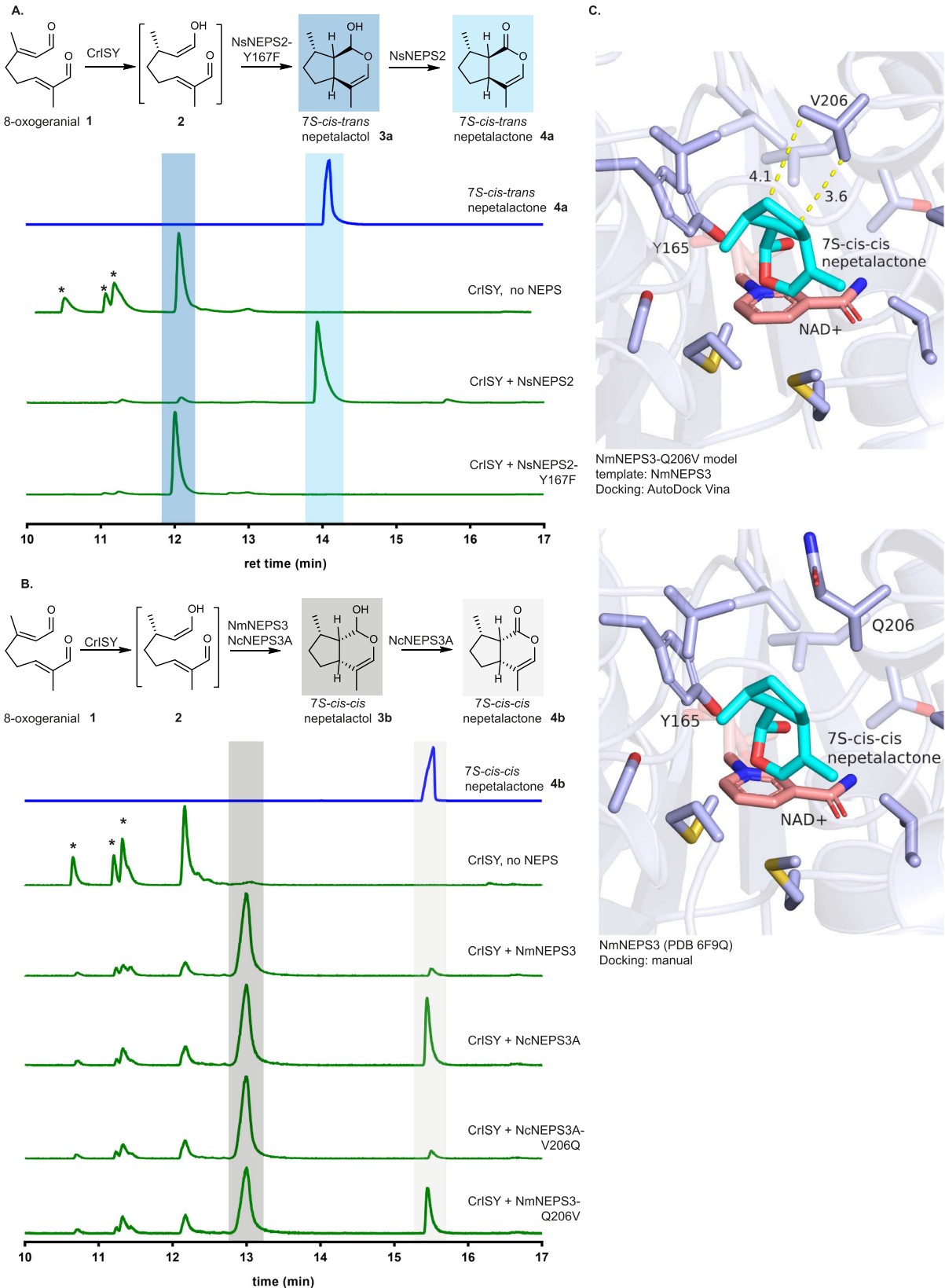

oxidation of **3** to nepetalactone **4** (Fig. 1D). We searched for mutations that would effectively decouple these cyclization and oxidative activities. Since NEPS belong to the NAD⁺-dependent short chain dehydrogenase family, mutation of the active site tyrosine residue required for oxidation in this class of enzymes could be a straightforward way to decouple cyclization and oxidative activity. Biochemical assays showed

that NsNEPS2 (*N. sibirica*) is a *cis-trans* cyclase and oxidase (Fig. 2A); when NsNEPS2 is incubated with 8-oxogeranial **1** and 7*S*-ISY, 7*S-cis-trans* nepetalactone **4a**, with no presence of iridodial **5** byproducts, is observed. It is important to note that in the absence of a cyclase, **1** and 7*S*-ISY yield the spontaneous cyclization product, 7*S-cis-trans* nepetalactol **3a**. However, numerous byproducts, most notably partially

**Fig. 2 | Decoupling of cyclization and dehydrogenation of 7S-cis-trans and 7S-cis-cis activities. A** *N. sibirica* NEPS2 (NsNEPS2) oxidation activity can be readily decoupled from the cyclization activity with catalytic Y167F amino acid substitution to produce 7S-cis-trans nepetalactol **3a**. Peaks marked with * represent iridodial **5** side products. **B** NcNEPS3A and NmNEPS3 are highly similar enzymes but only NcNEPS3A can oxidize 7S-cis-cis nepetalactol **3b**. NmNEPS3-Q206V mutant has restored oxidation activity and NcNEPS3A-V206Q has diminished oxidation activity

showing that this residue is directly involved in the binding of 7S-cis-cis nepetalactol **3b** to enable oxidation. **C** Molecular docking of 7S-cis-cis nepetalactol **3b** suggests that Q206 in NmNEPS3 could be negatively interacting with the cyclo-pentane ring in the nepetalactol and preventing binding. Highlighted parts of chromatograms represent the molecular structure highlighted with the same color. Results were repeated three times independently with similar results.

cyclized iridodials **5**, are observed in the spontaneous reaction (Fig. 1D), and therefore a *cis-trans* cyclase can be identified by the absence of these byproducts in the reaction mixture. Mutation of the active site tyrosine in NsNEPS2 (NsNEPS2-Y167F) yielded an enzyme that produces only *7S-cis-trans* nepetalactol **3a**, with no iridodial **5** byproducts, suggesting that cyclization activity was maintained while oxidation activity was eliminated (Fig. 2A). However, this did not prove to be a general strategy for decoupling of cyclization and oxidation activity in other NEPS, since mutation of the catalytic tyrosine residue in other NEPS enzymes yielded enzymes that were completely inactive or, surprisingly, unchanged in oxidation activity in NmNEPS1 and NsNEPS1B (Supplementary Fig. 3). In NmNEPS1 and NsNEPS1B there is clearly a compensatory, albeit subtle, mechanism that allows oxidation even in the absence of this tyrosine. One possibility is that as long as the substrate is positioned appropriately next to the cofactor, another residue- or water molecule- could serve as the base that would be required to facilitate this oxidation of the alcohol. However, with limited structural information on the NEPS it is not possible to pinpoint these subtle differences. Therefore, we focused on exploring changes to the rest of the binding pocket while leaving the catalytic residues intact.

To search for alternative mutations to decouple cyclization and oxidation, NmNEPS3 (*N. mussinii*), which produces 7S-cis-cis nepetalactol **3b** and only traces of the oxidized 7S-cis-cis nepetalactone **4b**, was compared with NcNEPS3A (*N. cataria* 93% amino acid identity, Supplementary Fig. 1), which carries out both cyclization and oxidation to **4b**[6] (Fig. 2B). This comparison suggested that residue 206 (Q in NmNEPS3, V in NcNEPS3A) could be responsible for the difference in oxidase activity. The mutant NmNEPS3-Q206V produces similar levels of **4b** as NcNEPS3A, and NcNEPS3A-V206Q produces only **3b** (Fig. 2B). Additional amino acid substitutions at residue 206 suggest that the polarity and size of the sidechain affect the oxidation of nepetalactol (Supplementary Fig. 4). Given th`e proximity of this residue to the binding site as evidenced by molecular docking (Fig. 2C), this residue may impact the exact orientation in which **3b** binds, thereby determining whether the lactol group is positioned such that it can be oxidized by the NAD⁺ cofactor. Thus we can control dehydrogenase activity but not impact cyclization in the 7S-cis-cis cyclase NEPS3 through a single substitution at residue 206.

Sequence comparison of these highly similar 7S-cis-cis cyclase enzymes, NmNEPS3 and NcNEPS3A, with that of other NEPS also revealed the identification of a variable loop region in the binding pocket (Supplementary Fig. 5A, B). Mutation of this loop in NcNEPS3A (from N150 to V162) with the corresponding residues from NmNEPS5, an enzyme known to produce 7S-cis-trans nepetalactone **4a**, resulted at successful swap of specificity from **4b** to **4a** (Supplementary Fig. 5C), showing that this loop plays a role in shaping the binding pocket and thereby setting the stereoselectivity of cyclization. However, when the same residues from NmNEPS4, a 7S-trans-cis **3c** cyclase, were introduced into NcNEPS3A, no 7S-trans-cis product was observed, albeit some **4a** was observed. We concluded that while the 150-162 loop is important in determining stereo-selectivity, it does not fully control it. Therefore, we needed to examine the role other regions of the binding pocket play in determining specificity.

To identify other regions responsible for controlling cyclization selectivity, we examined NmNEPS1 and NmNEPS4 due to the distinct

and seemingly co-dependent catalytic activities (Fig. 3A) and high amino acid identity of these two enzymes (81% amino acid identity). NmNEPS4 cyclizes 8-oxocitronellyl enol **2** to form 7S-trans-cis nepetalactol **3c**, which in the absence of a dehydrogenase spontaneously opens to form 7S-trans-cis iridodial **5c**[34,35]. NmNEPS1 has no cyclization activity, but NmNEPS1 is able to oxidize both **3c** and 7S-cis-trans nepetalactol **3a**. NmNEPS4 can only oxidize **3a** (Fig. 3A). Comparison of the binding pockets of these two enzymes revealed differences in eight residues with proximity to the binding pocket, six of which are located in the previously identified loop region in NEPS3 above (residues 152-164 in NmNEPS1 case), plus positions S104 and S198 in NmNEPS1 (equivalent to A105 and L199 in NmNEPS4) (Supplementary Fig. 6A, B). We replaced these 8 residues in NmNEPS1 (dehydrogenase activity only) with the residues found in NmNEPS4 (**3c** cyclase); gratifyingly, this resulted in a variant capable of generating **5c** iridodial, successfully demonstrating that this cyclization activity can be installed into a dehydrogenase (Fig. 3A, Supplementary Fig. 6C). A variety of mutant combinations showed that 5 mutations, 154SATA (in the loop region), along with L198, is the minimum requirement for formation of **5c** in the context of NmNEPS1. Based on the available NEPS crystal structures (NmNEPS3 and NsNEPS2), we hypothesized that the relatively bulky L199 residue in NmNEPS4 prevents the orientation of the **3c** relative to the NAD⁺ cofactor required for oxidation (Fig. 3C). As we predicted, reversing mutation of L198 back to serine resulted in a NmNEPS1 variant capable of both cyclizing and oxidizing to 7S-trans-cis nepetalactone **4c**, a catalytic activity not seen in the native NEPS. Extensive mutagenesis of the 154SATA region (Supplementary Fig. 7) as well as the L198 residue (Supplementary Fig. 8) impacted the ratio of **4c** and **4a** product ratios. In particular, 154SVTA mutant appears to improve the ratio towards **4c**, minimizing the amount of spontaneously formed **4a** (Fig. 3B). The 154SATA region enables the switch between *cis-trans* and *trans-cis* cyclization by shaping the binding pocket, though again, this switch must act in the context of two highly similar protein backgrounds.

## Discovery of trans-trans cyclase in N. sibirica

The 7S-trans-trans nepetalactol **3d** or nepetalactone **4d** isomers were not observed in any of these mutagenesis experiments, prompting us to search for the naturally occurring cyclase that produces this stereoisomer. Based on achiral GC-MS analysis, we observed a compound that appeared to have **4d**, in the phylogenetically distant *N. sibirica*, prompting us to search for cyclases with 7S-trans-trans stereo-selectivity in this plant. We identified and functionally characterized all *ISY*, *NEPS*, and *MLPL* orthologues in the *N. sibirica* leaves transcriptome (Supplementary Tables 2–3). First, orthologues of *ISY* and the closely related progesterone 5β-reductase (*P5βR*) from *N. sibirica* were assayed and analyzed in a GC chiral column, revealing that all of these enzymes reduced 8-oxogeranial **1** with 7 S specificity at C7 (Fig. 4A), the same as ISY enzymes from *C. roseus* as well as *N. cataria* and *N. mussinii*. A relative ISY from the Lamiaceae species *Lamium album* (LaISY) was identified and used to generate the products with 7 R stereochemistry. Thus, we initially tested the activity of all putative *N. sibirica* cyclases with the *N. sibirica* iridoid synthase in search of 7S-trans-trans nepetalactone activity.

As described above, we identified, characterized and subjected to crystallography analysis the NEPS orthologue NsNEPS2 as a 7S-cis-trans nepetalactone **4a** synthase. Additional NEPS orthologues NsNEPSL,

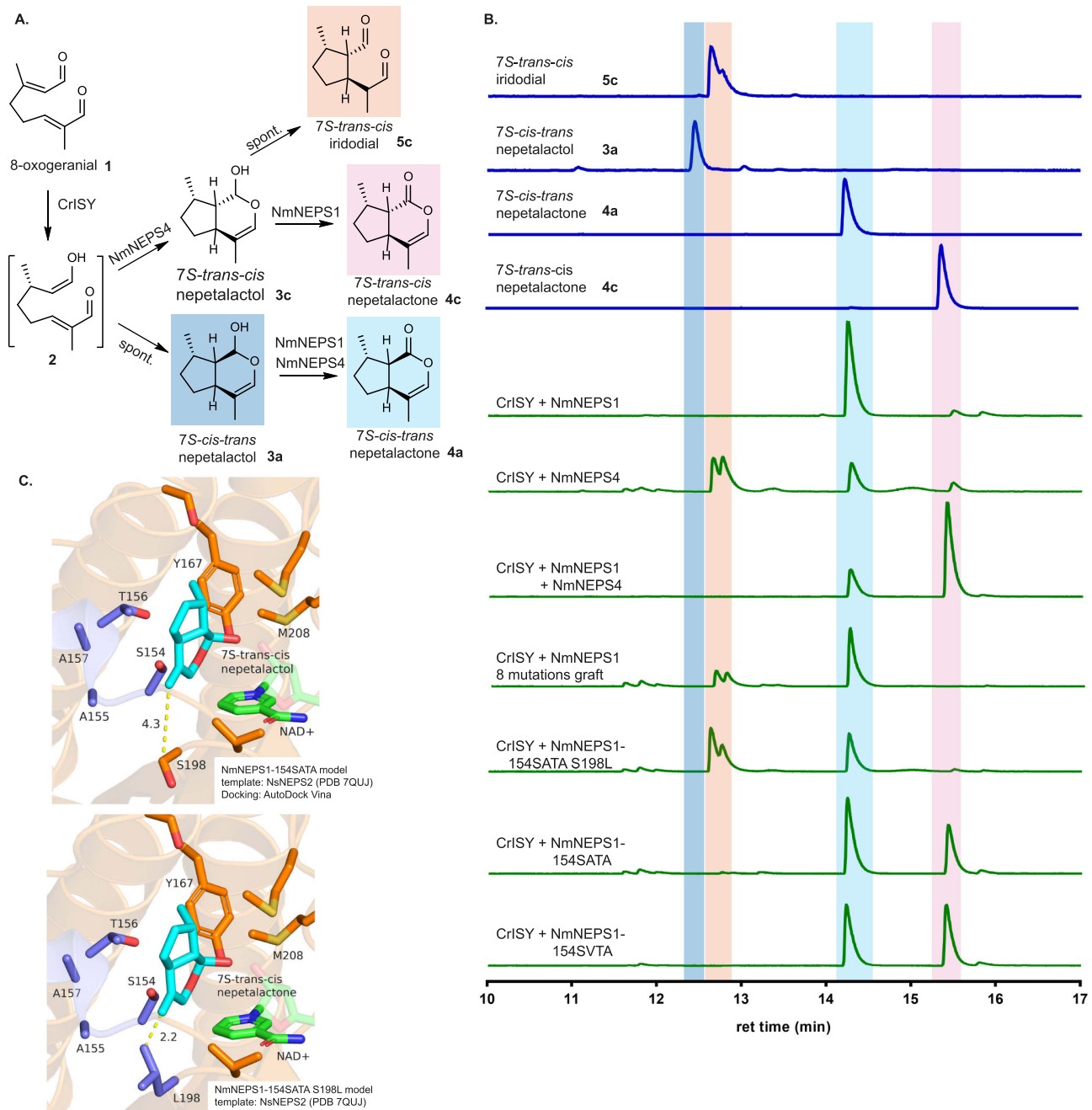

**Fig. 3 | Decoupling of cyclization and dehydrogenation of 7S-trans-cis activity.**
**A** Existing enzymes have only partial roles in cyclization (NmNEPS4) and oxidation (NmNEPS1) of *7S-trans-cis* nepetalactol **3c**. **B** NmNEPS1 was engineered to perform both cyclization and oxidation to form *7S-trans-cis* nepetalactone **4c**. The best variant found (NmNEPS1-154SVTA) is able to produce significantly more **4c** than either NmNEPS1 or NmNEPS4 alone. In addition, L199 residue in NmNEPS4 appears to be involved in destabilizing the *7S-trans-cis* nepetalactol **3c**. **C** Molecular docking of **3c** showing the 154SATA loop and L198 suggesting these residues might clash with the C4 methyl group in **3c**, preventing proper positioning for oxidation. Highlighted parts of chromatograms represent the molecular structure highlighted with the same color. Results were repeated three times independently with similar results.

NsNEPS1A, NsNEPS1B, NsNEPS4A, and NsNEPS4B were also identified (Fig. 4B, Supplementary Figs. 1–2). No orthologs of NEPS3 (*7S-cis-cis*) or NEPS5 (*7S-cis-trans*) were found. RNA-seq data suggests that NsNEPS1A, NsNEPS1B, NsNEPS4B are the most highly expressed genes, with NsNEPS1A being the highest (Fig. 4C).

Enzyme assays with 8-oxogeranial and NsISY show that NsNEPS1A, NsNEPS1B, NsNEPS2 and NsNEPSL all oxidize *7S-cis-trans* nepetalactol **3a** to *7S-cis-trans* nepetalactone **4a** (Fig. 4D, E). NEPS4 (NsNEPS4A and NsNEPS4B) lacked any activity under these conditions. NsMLPL1 appeared to be a **3a** cyclase while the other MLPL homologs did not show activity towards 8-oxocitronellyl enol **2** (Fig. 4D, E). However, NsNEPS1B produced an unprecedented stereo-chemical profile, with all four of the *7S* nepetalactone isomers, including *7S-trans-trans* nepetalactone **4d**, being observed in the reaction mixture (Fig. 4D). Curiously, although **4d** activity was achieved, the ratio in which it was produced was not consistent with the profile observed in *N. sibirica*. Upon reviewing the literature, we found a report demonstrating the existence of *7R-trans-trans* nepetalactone **4d'** in *N. elliptica*[36]. Although NsISY is in fact *7S* specific, we decided to rely on the *7R* specific LaISY to test this hypothesis.

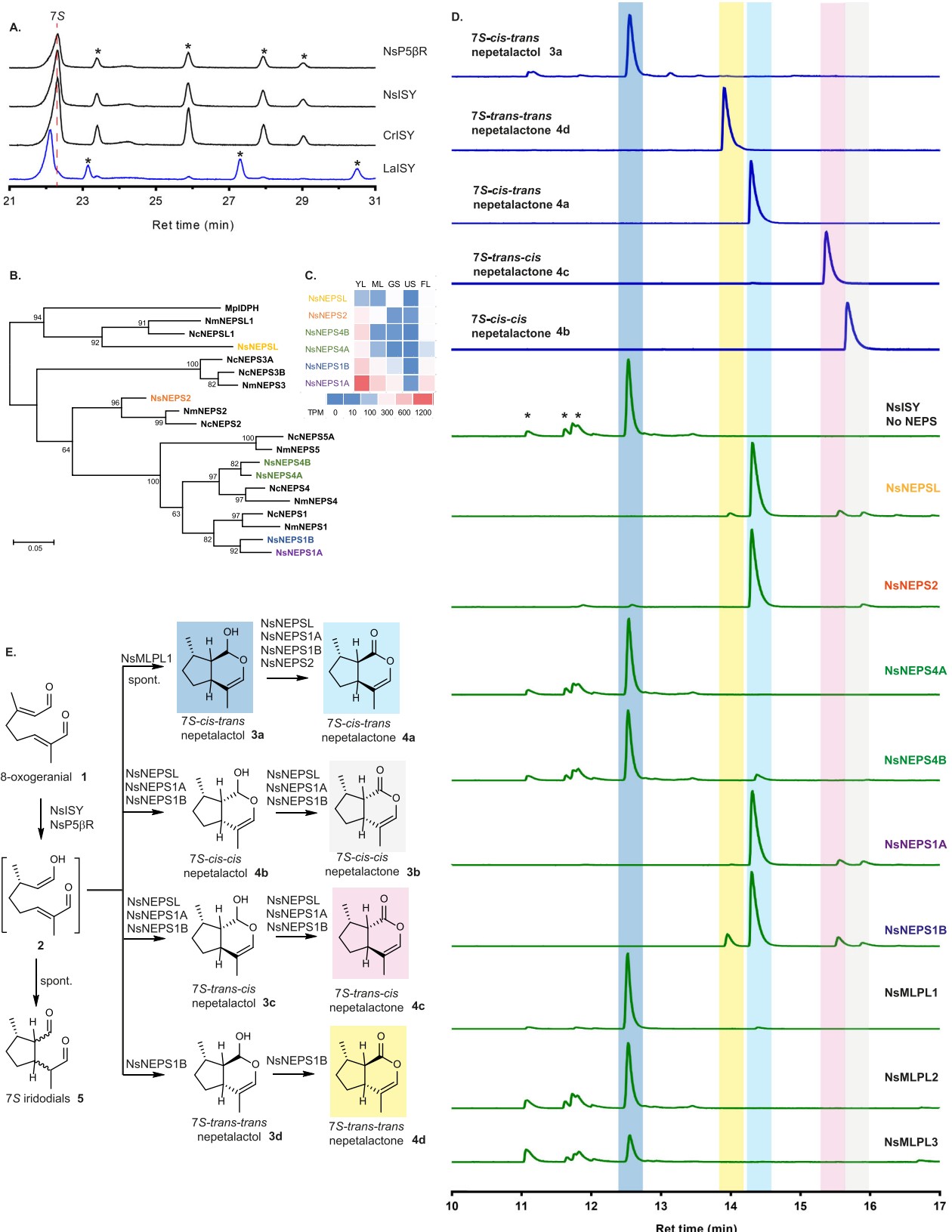

## Reactions with 7R-ISY suggest the role of NEPS in *N. sibirica*

Given the prevalence of iridoids with 7 *R* stereochemistry outside of the *Nepeta* genus, we tested the cyclization and oxidation activity of the NEPS in combination with 7*R*-ISY to determine whether NEPS could also catalyze the stereo-selective cyclization/oxidation of 7 *R* iridoid scaffolds. 7*R*-ISY from the plant *Lamium album* (LaISY) was used as the

7*R*-ISY (Figs. 4A, 5A). NsNEPS1B, which can produce 7*S-trans-trans* nepetalactone, produced both 7*R-trans-trans* nepetalactone **4d'** and 7*R-cis-trans* nepetalactone **4a'** when assayed with LaISY and 8-oxogeranial **1** (Fig. 5B). The most highly expressed NEPS in *N. sibirica*, NsNEPS1A made exclusively 7*S-cis-trans* nepetalactone **4a** when assayed with 7*S*-ISY (Fig. 4D), but in combination with LaISY, NsNEPS1A

**Fig. 4 | *Nepeta sibirica* ISY, NEPS and MLPL screening. A** Chiral GC-MS traces show that both NsISY and NsP5ßR have 7*S*-ISY activity, as evidenced by the alignment of the products (7*S*-*cis-trans* nepetalactol **3a** and iridodials **5**) with those of the 7*S*-specific CrISY and not the 7*R*-specific LaISY. The red dashed line represents the 7*S*-*cis-trans* nepetalactol **3a** peak and the peaks marked with * represent iridodial **5** side products. **B** Phylogenetic tree of all NEPS from *N. sibirica* found and their relationship with *N. mussinii*, and *N. cataria*. **C** Relative expression of NEPS in *N. sibirica* shows that young leaves tissue contains the highest expression profile and NsNEPS1A is the most highly expressed of the NEPS. **D** Enzymatic activity of *N. sibirica* enzymes. NsNEPS as well as NsMLPL identified in combination with identified NsISY shows the various nepetalactone profiles obtained. Most notably, NsNEPS1B produces a significant amount of 7*S*-*trans-trans* nepetalactone **4d**. **E** Scheme of activities found by combining NsISY and various NsNEPS and NsMLPL enzymes. Highlighted parts of chromatograms in part **D** represent the molecular structures highlighted with the same color in part **E**. NEPS names in parts **B**, **C**, and **D**, are color coded. Results were repeated twice independently with similar results.

produced 7*R*-*trans-trans* nepetalactone **4d'** as the main product, along with lower amounts 7*R*-*cis-trans* nepetalactone **4a'** (Fig. 5B). Efforts to decouple cyclization and dehydrogenation of these enzymes via catalytic mutation (Supplementary Fig. 3) or via binding pocket redesign (Supplementary Fig. 9) did not yield enzymes with substantially different catalytic activities albeit changes in the ratio of nepetalactones were observed.

These enzymatic results led us to re-inspect the nepetalactone profile of *N. sibirica* using a combination of chiral GC column (Fig. 5B), chemical epimerization[34] of 7*S*-*cis-cis* nepetalactone **4b** derived from *N. mussinii* and putative *N. sibirica trans-trans* nepetalactone (Fig. 5C; Supplementary Figs. 10, and 14–17), and circular dichroism analysis of the resulting enantiomers (Fig. 5D). Careful analysis lead us to confirm that the leaves of *N. sibirica* contained 7*R*-*trans-trans* nepetalactone **4d'** and not 7*S*-*trans-trans* nepetalactone **4d**, as we had initially assumed upon initial inspection of the leaf profile with a standard achiral GC column. Low amounts of 7*S* and 7*R*-*cis-trans* nepetalactones (**4a** and **4a'**, respectively) were also observed in *N. sibirica* leaves. This result is surprising given that only 7*S*-specific ISY enzymes have been identified in *N. sibirica*, and throughout the *Nepeta* genus. We also tested all *N. sibirica* NEPS for ISY activity, but observed no activity for this reaction. Assays of 8-oxoneral **6** (which would result from neryl-pyrophosphate intermediate instead of geranyl pyrophosphate[37]) with NsISY or NsP5ßR also failed to yield 7*R* products (Supplementary Fig. 11). It is likely that the *N. sibirica* 7*R*-ISY is not a homolog of ISY or NEPS and therefore identification of this enzyme will require a more intensive investigation. In summary, the high expression and in vitro activity of NsNEPS1A strongly suggests that this enzyme is a bona fide *trans-trans* cyclase and oxidase that is responsible for the synthesis of 7*R*-*trans-trans* nepetalactone **4d'** in planta.

## Combinatorial biosynthesis of nepetalactol and nepetalactone isomers

Enzymes that can catalyze formation of the four 7*S* stereoisomers were now in hand: (i) 7*S*-*cis-trans* (7*S*,4a*S*,7a*R*) with NsNEPS2 (lactone **4a**) and NsNEPS2-Y167F (lactol **3a**); (ii) 7*S*-*cis-cis* (7*S*,4a*R*,7a*S*) **4b** with NmNEPS3-Q206V or NcNEPS3A; (iii) 7*S*-*trans-cis* (7*S*,4a*S*,7a*S*) **4c** with NmNEPS1-155SVTA; (iv) 7*S*-*trans-trans* (7*S*,4a*R*,7a*R*) **4d** with NsNEPS1B (albeit as a minor component). Furthermore we showed that we can access 7*R*-*trans-trans* (7*R*,4a*S*,7a*S*) nepetalactone **4d'** using 7*R*-ISY and NsNEPS1A/B. We set out to determine whether we could generate the other 7*R* stereoisomers using combinations of NEPS and 7*R*-ISY to gain access to diverse iridoid stereoisomers.

Previous work had established that 7*R*-ISY produces 7*R*-*cis-trans* (7*R*,4a*R*,7a*S*) nepetalactol **3a'** through spontaneous cyclization, though numerous iridodial byproducts are observed. NsNEPS2, 7*S*-*cis-trans* nepetalactone **4a** synthase when assayed with 7*S*-ISY, yielded 7*R*-*cis-trans* nepetalactone **4a'** when assayed with 7*R*-ISY. Correspondingly, NsNEPS2-Y167F yielded 7*R*-*cis-trans* nepetalactol **3a'** (Fig. 6A, B). Assays of the remaining NEPS with 7*R*-ISY revealed that: NsNEPS4B produced trace levels of **4a'** while no activity was observed for NsNEPS4A. NsNEPSL had **4a'** activity though low levels of 7*R*-*trans-trans* nepetalactone **4d** were also observed (Supplementary Fig. 12). NEPS3-Q206V produced 7*S*-*cis-cis* (7*S*,4a*R*,7a*S*)

nepetalactone **4b** (Fig. 2B), and as a result we hypothesized that this enzyme could also generate the enantiomer 7*R*-*cis-cis* nepetalactone **4b'**. However, NmNEPS3-Q206V did not produce **4b'** as expected but **3a'** (Fig. 6B). Analogously, we attempted to use NmNEPS1-155SVTA (Fig. 3B), which produced 7*S*-*trans-cis* **4c**, but instead we observed 7*R*-*trans-trans* (7*R*,4a*S*,7a*S*) **4d'** nepetalactone when assayed with 7*R*-ISY (Fig. 6D). Moreover, the fraction of 7*R*-*trans-trans* nepetalactone **4d'** relative to 7*R*-*cis-trans* nepetalactone **4a'** when NmNEPS1-154SVTA is employed appears to be larger than NsNEPS1A or NsNEPS1B. Screening of 7*R*-ISY with all individual NEPS and MLP enzymes failed to produce an enzyme system that yielded 7*R*-*cis-cis* **4b'** or 7*R*-*trans-cis* **4c'**. Finally, we explored the possibility of using the more promiscuous NsNEPS1B in combination with *N. sibirica* MLPL enzymes. Although NsNEPS1B does not produce 7*R*-*cis-cis* nepetalactone, when combined with NsMLPL2, 7*R*-*cis-cis* **4b'** appears as one of the secondary products (Fig. 6C). Importantly, although enzymatic yields of **4b'** could be further optimized, **4b'** can be readily obtained from enzymatically derived 7*R*-*trans-trans* nepetalactone **4d'** via chemical epimerization (Fig. 5C). Therefore, 7 out of the 8 possible nepetalactone stereoisomers have been accessed in vitro using a combination of native and mutant NEPS/MLPLs along with the two stereoselective ISY enzymes. In summary, the active site of NsNEPS2 can accommodate formation of both enantiomers, 7*R*-*cis-trans* **4a** and 7*S*-*cis-trans* **4a'** which have two unique 4a-7a bridge configurations (4a*R*,7a*S* vs. 4a*S*,7a*R*, respectively). Similarly, NsNEPS1B can also accommodate formation of both enantiomers, 7*S*-*trans-trans* **4d** and 7*R*-*trans-trans* **4d'** though formation of **4d'** is clearly favored. In contrast, NmNEPS3-Q206V and NmNEPS1-154SVTA appear to be highly specific for the bridgehead carbon stereochemistry (4a*R*,7a*S* and 4a*S*,7a*S*, respectively) regardless of the C7 methyl group configuration (Fig. 6E).

Finally, we surveyed additional *N. mussinii*, *N. cataria* and *H. officinalis* selected NEPS and MLPL enzymes. Surprisingly, *H. officinalis*, a mint family plant that does not accumulate nepetalactones[25], possesses NEPS-like (NEPSL) enzymes that are able to oxidize both 7*S* and 7*R cis-trans* nepetalactol (**3a** and **3a'**, respectively) to form the corresponding nepetalactones (Supplementary Fig. 13). Moreover, NEPSL enzymes of *N. mussinii*, *N. cataria* also catalyze the same 7*S* and 7*R cis-trans* oxidation. In addition, *N. cataria* and *N. mussinii* NEPS5 enzymes, which do not have cyclization activity for 7*S* substrates do catalyze cyclization of 7*R*-*trans-trans* nepetalactol **3d'** and its subsequent oxidation to nepetalactone **4d'**. Taken together, these results suggest that nepetalactol oxidation activity is more widespread in the mint family than previously thought and that it is perhaps the context of the presence or absence of an ISY-like enzymatic activity that controls whether the plant accumulates nepetalactones, or whether it is of 7*S* or 7*R* configuration.

The iridoid scaffold has potential applications as an insect repellant but it is difficult to chemically synthesize in a stereo-selective manner[38–40]. Here we report the discovery of NEPS enzymes, both naturally occurring and engineered, capable of stereo-divergent enzymatic synthesis of 7 iridoid scaffold stereoisomers. Gene discovery efforts in a phylogenetically distant *Nepeta* species led to the discovery of a NEPS with previously uncharacterized cyclase activity. Structural characterization and mutagenesis demonstrated that cyclization and oxidation activity could be decoupled in the NEPS, resulting

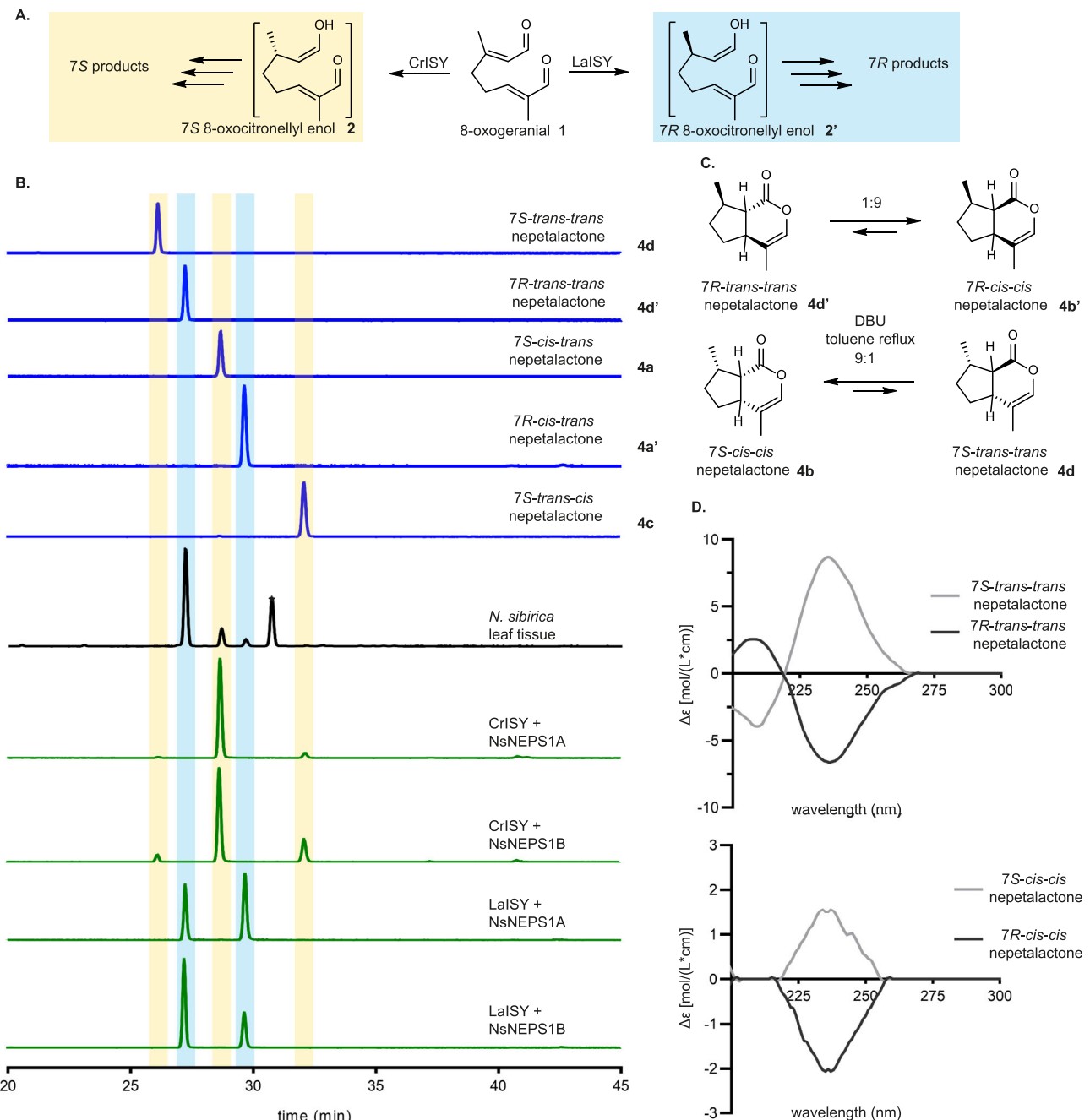

**Fig. 5 | *Nepeta sibirica* produces both *7R* and *7S* iridoids. A** The 7 S specific ISY comes from *C. roseus* while the *Lamium album* iridoid synthase (LaISY) yields the 7 R stereoisomer **2'**. **B** GC-MS traces using a chiral column capable of separating 7 S (yellow highlight) and 7 R (light blue highlight) enantiomers. Peak indicated with * in *N. sibirica* leaf trace corresponds to germacrene. Surprisingly, *N. sibirica* leaves produce largely 7 R nepetalactones with 7 S nepetalactones as a minor product. When NsNEPS1A and NsNEPS1B are tested with LaISY the resulting products match those found in the plant. **C** DBU based epimerization scheme to confirm the C7 stereochemistry of *N. sibirica trans-trans* nepetalactone. Using previously characterized *7S-cis-cis* nepetalactone **4b** from *N. mussinii* and converting it to *7S-trans-trans* **4d** it is possible to compare both molecules using chiral column GC-MS (part B). **D** Circular dichroism spectra was used to distinguish the nepetalactone enantiomers. Results were repeated twice independently with similar results.

in more versatile biocatalysts. Moreover, structural elements that controlled the stereo-selectivity of the cyclization were identified in the NEPS proteins, providing mechanistic insight into this unusual class of cyclases. The discovery of a group of naturally occurring orthologues with varying product selectivity served as a starting point for modulating oxidative and cyclization selectivity. The impact of the engineered mutations was often dependent on the specific protein. While this precluded formulating a generalizable set of design rules for engineering these enzymes, we nevertheless could make some hypotheses regarding the mechanistic basis of how these mutations

modulated cyclization and oxidation activities. Finally, we could productively combine cyclases to lead to a non-naturally occurring stereoisomer.

Plants have evolved an expansive chemical repertoire comprising $10^5$-$10^6$ known compounds[1-3]. Our ability to understand and harness this chemical diversity has wide ranging applications in agricultural pest management, crop domestication, food industry, medicines, and specialty chemical production. Thus, identification and engineering of the biosynthesis genes that are responsible for the synthesis of these molecules presents opportunities for

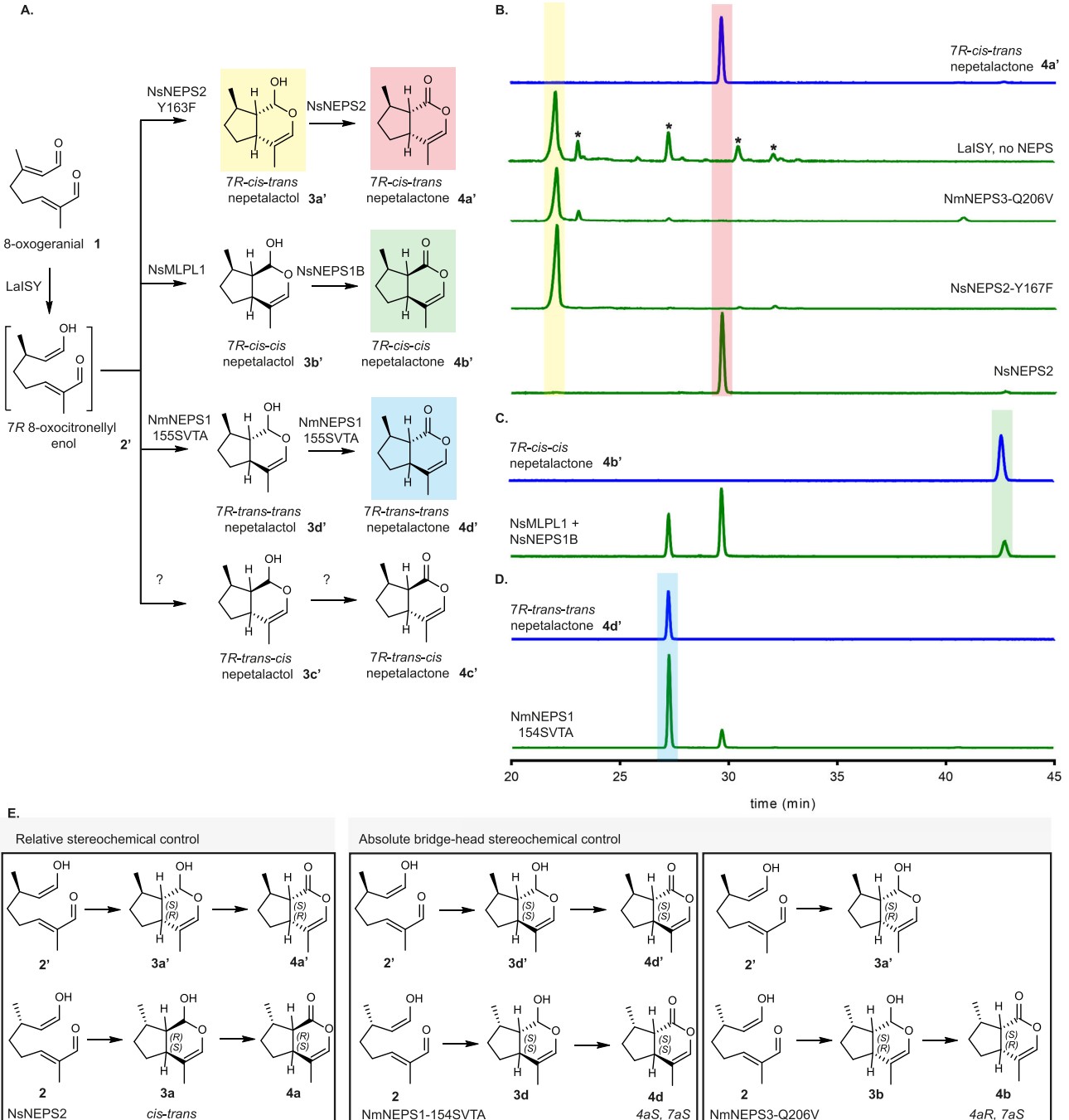

**Fig. 6 | Combinatorial biosynthesis for the production of 7R nepetalactones.**
**A** Various NEPS and MLPL enzymes were combined with LaISY in order to access 7R nepetalactones. No NEPS combination produced *7R-trans-cis* nepetalactone **4c'**. **B** Similarly to *7S-cis-trans* nepetalactone **4a** (Fig. 2A), NsNEPS2 and NsNEPS-Y167F are capable of effectively producing *7R-cis-trans* nepetalactone **4a'** and *7R-cis-trans* nepetalactol **3a'**, respectively. Peaks marked with * represent iridodial **5** side products. **C** A combination of NsMLPL1 and NsNEPS1B generated *7R-cis-cis* nepetalactone **4b'** as a minor product. **D** Our best *7S-trans-cis* nepetalactone producer,

NmNEPS1-154SVTA is capable of high production of *7R-trans-trans* nepetalactone **4d'**. Highlighted parts of chromatograms represent the molecular structure highlighted with the same color. **E** Depiction of how NEPS have different ways to control the stereochemistry of cyclization. NsNEPS2 is able to catalyze cyclization of enantiomers yielding *cis-trans* stereochemistry for both 7S and 7R ISY. NmNEPS-154SVTA and NmNEPS3-206V appear to have specificity for the absolute configuration of the bridgehead carbons, and the orientation of the C7 methyl group does not change it. Results were repeated three times independently with similar results.

production of high value molecules using metabolic engineering and synthetic biology approaches. To date, *7S-cis-trans* nepetalactol **3a**, which is also formed spontaneously in the absence of any cyclase, albeit in greatly reduced yield, has been the main focus of microbial engineering efforts[29–33]. The enzymes reported in this study can be utilized in these metabolic engineering systems to expand the iridoid-derived natural product space and potentially generate natural product derivatives.

## Methods

### Chemicals

8-oxogeranial (CAS# 80054-40-6) & 7S-cis-trans nepetalactol (CAS# 109215-55-6) from Toronto Research Chemicals. 7S-cis-trans and 7S-trans-cis nepetalactone were isolated from *Nepeta fassinii* "Six Hills Giant" (Burncoose Nurseries, Gwennap, UK) and from catnip oil, respectively using flash silica gel chromatography[41]. 7S-cis-cis nepetalactone was isolated from *Nepeta mussinii* "Snowflake"

(Burncoose Nurseries, Gwennap, UK) also using using flash silica gel chromatography[20]. 8-oxonerol was synthesized[19] by hydrolysis of neryl acetate (Sigma-Aldrich) as an initial substrate. To synthesize the dialdehyde (8-oxoneral), 8-oxonerol (25 mmole) was added to a mixture of $MnO_2$ (175 mmole) in dichloromethane (50 mL) and stirred at room temperature for 16 h. The mixture was filtered through celite, concentrated *in vacuo*, and 8-oxoneral was purified via silica flash chromatography using a gradient from 0 to 50% EtOAc in hexane. 7R-trans-trans nepetalactone was isolated from *N. sibirica* (Beth Chatto, Colchester, UK) following the method for *N. racemosa* in Liblikas, et al.[34].

## Cloning

Primers and synthetic genes were purchased from Integrated DNA Technologies (IDT). To clone NEPS, MLPL and ISY genes from *N. sibirica* cDNA library, primers (see Supplementary Table 2) contained overhangs with homology to the pOPINF vector (Addgene plasmid #26042)[42]. Putative genes from *N. sibirica* as well as all NEPS variants generated, were cloned into the pOPINF vector using the commercial kit Infusion HD (Clontech). Extracted RNA (see transcriptome section) was DNAse treated with TURBO DNA-free kit (Thermo Fischer) and used to build cDNA library using SuperScript IV VILO (Invitrogen) master mix. Mutations to NEPS genes were done with primers containing the sequence changes and overhangs compatible with Infusion HD cloning (Supplementary Table 6). The nucleotide sequence corresponding to *Lamium album* Iridoid Synthase was obtained by retrieving the sequences of the Lamiaceae ISY orthogroup from previously published data[43]. DNA fragments with overhangs with homology to pQLinkH were ordered from ThermoFisher GeneArt Strings service, and cloned into the pQLinkH[44] vector (Addgene plasmid # 13667) using the commercial kit Infusion HD (Clontech).

## Enzyme expression and purification

Unless otherwise noted, genes cloned into pOPINF vectors (NEPS, MLPL, ISY) and pQLinkH (LaISY) vectors were transformed into SoluBL21 *E. coli* strain for protein expression and carbenicillin (50 µg/mL) was used for selection. Single colonies grown in LB agar media were picked and grown overnight in 10 mL LB media at 37 °C. Next day, 1 mL of the overnight cultures were used to inoculate 100 mL of 2xYT media. The fresh cultures were grown at 37 °C until $OD_{600}$ of 0.6-0.8 followed by induction with 250 µM IPTG and incubation at 18 °C for 16–18 h. For purification, the cells were harvested by centrifugation (10 min at 3200 g), resuspended in 10 mL of binding buffer (50 mM tris-HCl pH 8, 50 mM glycine, 500 mM Sodium Chloride, 20 mM imidazole, 5% v/v glycerol, pH 8) containing in addition 0.2 mg/mL Lysozyme and EDTA free protease inhibitor cocktail (Roche cOmplete[TM]) and incubated for 30 min on ice. Cells were lysed by sonication using a Sonics Vibra Cell at 40% amplitude, 3 s ON, 2 s OFF, 4 min total. The crude lysates were centrifuged at 35,000 g for 15 min and the cleared lysates were incubated with 250 uL Ni-NTA agarose beads (Qiagen) for 30 min at 4 °C. Next, the beads were sedimented by centrifugation at 1000 g for 2 min and washed three times with binding buffer before eluting the proteins with elution buffer (50 mM tris-HCl pH 8, 50 mM glycine, 500 mM Sodium Chloride, 250 mM imidazole, 5% v/v glycerol, pH 8). Proteins were then buffer exchanged into assay buffer (50 mM HEPES pH 8) using Amicon Ultra centrifuge filters and stored at −80 °C until assays were done. A representative SDS-PAGE gel of proteins purified for Fig. 3B and Supplementary Fig. 6 is shown in Supplementary Fig. 18.

For crystallization trials NsNEPS2 gene sequence was codon optimized for *E. coli* expression and purchased from Integrated DNA Technologies. The synthetic gene was cloned into pOPINF expression vector with an N-terminal his tag and 3 C protease site (MAHHHHHHSSGLEVLFQGP) using In-fusion cloning kit (Takara).

*E. coli* SoluBL21 cells were transformed with pOPINF-NsNEPS2. Selection with carbenicillin (50 µg/mL) was kept throughout this procedure. A single colony grown in LB agar medium was picked and grown overnight in 10 mL LB media at 37 °C. Next day, the overnight culture was used in a 1:100 ratio to inoculate 100 mL seed culture. Finally, the seed culture was used in 1:100 ratio to inoculate 2 x 1 L cultures in 2xYT medium. Cultures were grown at 37 °C until $OD_{600}$ of 0.6 followed by 30 min growth at 18 °C, induction with 250 µM IPTG and further growth at 18 °C for 16–18 h. For purification, the cells were cooled down on ice, harvested by centrifugation for 15 min at 3200 g at 4 °C, resuspended in 50 mL/L culture of binding buffer (50 mM tris-HCl pH 8, 50 mM glycine, 500 mM Sodium Chloride, 20 mM imidazole, 5% v/v glycerol, pH 8) containing in addition 0.2 mg/mL Lysozyme and EDTA free protease inhibitor cocktail (Roche cOmplete[TM]). After incubation on ice for 30 min, the cells were lysed by sonication using a Sonics Vibra Cell at 40% amplitude, 3 s ON, 2 s OFF, 4 min total. Next, the lysate was centrifuged at 35,000 g for 15 min and the cleared lysate was applied to Akta pure system for purification. The purification was done in three steps. First the lysate as applied to a 5 mL HiTrap column (GE) to bind NEPS2 and eluted with a buffer containing 500 mM imidazole. Next the eluted protein was applied to a HiLoad 16/600 Superdex 200 pg (GE) with 20 mM HEPES pH 7.5, 150 mM NaCl as running buffer for size exclusion purification. Finally, the protein was concentrated and buffer exchanged into Low salt buffer (20 mM Tris-HCl pH 7.5, 20 mM NaCl) for ionic exchange purification. The protein was applied to a Mono Q 5/50 GL (GE) column and eluted with High salt buffer (20 mM Tris-HCl pH 7.5, 1000 mM NaCl) using a linear gradient. The protein was concentrated and buffer exchanged back into Low salt buffer using Amicon 10 kDa MWCO centrifuge filters and the final concentration of protein was 9.6 mg/mL.

## Enzyme assays and extraction

End-point enzymatic assays where 8-oxogeranial is combined with ISY and NEPS/MLPL were done under the following conditions unless otherwise noted: 50 mM HEPES pH 8, 0.5 mM 8-oxogeranial, 1 mM NADPH, 5 mM NAD + , 0.5 µM for CrISY or 2 µM for LaISY, and 10 µM of NEPS or MLPL in 100 µL total volume. The assays were set up in 1.5 mL Eppendorf tubes and incubated for 3 h at 30 °C at 700 rpm. It is important to ensure that NEPS protein stocks do not warm to room temperature unless NAD+ is present, otherwise, protein precipitation can occur. After reaction, 10 uL of 1 mM Camphor in acetonitrile was added to the reaction products as internal standard and the reaction was then extracted with 100 µL of ethyl acetate. The mixture was vortexed for 2 min and centrifuged at 15,000 g for 5 min before collecting the ethyl acetate layer for GC-MS analysis.

## GC-MS analysis

For analysis of nepetalactones on achiral column, samples were injected into a Thermo Trace 1310 GC-MS with an ISQ LT mass selective detector and a CTC Analytics PAL GC-xt autosampler. The samples (1 µL) were injected at inlet temperature of 230 °C in 1:10 ratio split mode into a Zebron ZB5-HT-INFERNO (ID = 0.25 mm; $L$ = 30 m; film thickness = 0.1 µm). The mobile phase was helium at constant flow of 1.1 mL/min. The oven temperature ramp was as follows: starting at 80 °C, hold 5 min, ramp at 2.5 °C/min to 110 °C, ramp at 50 °C/min to 280 °C, hold 4 min. The transfer line to MS was kept at 280 °C and the MS source was at 250 °C. After a solvent delay of 4 min, mass range of 50-330 amu was collected at a fragmentation energy of 70 eV.

For analysis of nepetalactones on chiral column, samples were injected into an Agilent 8890 GC system with an Agilent 5977B mass selective detector and an Agilent 7693 A autosampler. The samples (1 µL) were injected at inlet temperature of 220 °C in 1:10 ratio split

mode into a Supelco Beta DEX 225 column (ID = 0.25 mm; $L$ = 30 m; film thickness = 0.25 μm). The mobile phase was helium at constant flow of 1.1 mL/min. The oven temperature ramp was as follows: starting at 80 °C, hold 3 min, ramp at 10 °C/min to 120 °C, hold 45 min, ramp at 10 °C/min to 200 °C, hold 2 min. The transfer line to MS was kept at 220 °C and the MS source was at 230 °C. After a solvent delay of 10 min, mass range of 50–350 amu was collected at a fragmentation energy of 70 eV.

## Phylogenetic tree and sequence alignments

Protein sequence alignment was constructed in Geneious Prime (2019.1.1) software using the ClustalW (v2.1) (Cost matrix = BLOSUM, gap open cost = 10, gap extend cost = 0.1)[45] option. Phylogenetic tree was inferred with the Maximum Likelihood method and Poisson correction model[46] tree was constructed in MEGAX 10.2.2 software[47] using the Bootstrap test with 1000 replications[48], and a discrete Gamma distribution (5 categories).

## Circular dichroism

For circular dichroism measurements, nepetalactone isomer samples were diluted in methanol at 0.1 mg/mL. Samples at room temperature (20 °C) were placed in a cuvette with cell length of 1 mm. Spectra were measured on a Jasco J-810 spectro-polarimeter from 400–190 nm in 1 nm steps at a scanning speed of 100 nm/min.

## NsNEPS2 crystallization

NsNEPS2 was purified as described above. NsNEPS2 crystals were formed using the sitting drop method. The precipitant solution was 0.2 M Sodium acetate with 11.5% w/v PEG 4000. Droplet conditions were: 0.6 μL total volume, 200 nL protein + 400 nL precipitant, 10x molar excess NAD$^+$ in the protein stock. Prior to harvesting, crystals were soaked for 5 min in the precipitant solution containing 1.66 mM NAD+, 6.66 mM cis-trans-nepetalactone and 25% ethylene glycol as cryo-protectant. Data collected contained NAD+ bound but not nepetalactone.

## NsNEPS2 crystals data collection and structure determination

X-ray data sets were recorded on the 10SA (PX II) beamline at the Paul Scherrer Institute (Villigen, Switzerland) at wavelength of 1.0 Å using a Dectris Eiger 16 M detector with the crystals maintained at 100 K by a cryocooler. Diffraction data were integrated using XDS[49] (BUILT = 20220220) and scaled and merged using AIMLESS[50] (0.7.7); data collection statistics are summarized in Supplementary Table 1. Initially the NsNEPS2 data set was automatically processed at the beamline to 1.85 Å resolution and a structure solution was automatically obtained by molecular replacement using pdb 6F9Q as template. The map was of sufficient quality to enable 90% of the residues expected for a NsNEPS2 homodimer to be automatically fitted using Phenix autobuild[51] (1.19.2). The model was finalized by manual rebuilding in COOT[52] (0.9.6) and refined using in Phenix refine[53] (1.19.2) and is deposited as PDB entry 7QUJ.

## Protein crystal modeling and molecular docking

Homology models were built using the Swiss-Model server[54]. When models were based on NmNEPS3 crystal structure, PDB 6F9Q was used as template. When models were based on NsNEPS2 (now PDB 7QUJ), the crystal structure map was introduced manually. Protein crystal structures/models were visualized in Pymol (2.3.2). Each model figure indicates what protein was used as template. NmNEPS1-154SATA model has a Global Model Quality Estimate (GMQE)[55] value of 0.83 and NmNEPS3-Q206V model had a QMEAN value of 0.94. Molecular docking of nepetalactones into NEPS was performed using AutoDock Vina (1.1.2)[56]. Ligand 3D structures were generated in MarvinSketch (19.19.0) and AutoDockTools (1.5.6) was used to prepare the ligands and receptor molecules for docking.

## *N. sibirica* genes transcriptome assembly and gene expression abundance estimation

Plants were grown in a greenhouse exposed to the temperature and humidity of the environment in Norwich, UK. Plants were cut back and young leaf tissue collected after 2 weeks. RNA was isolated using a CTAB method[57] prior to DNase treatment using the Turbo DNA-Free Kit (ThermoFisher Scientific, Waltham, MA, Cat #AM1907), and checked for quality with a Nanodrop and a TapeStation (Aligent, Santa Clara, CA). RNA was then used to construct an Illumina TruSeq Stranded mRNA library (Illumina, San Diego, CA) and sequenced to 150 nt on an Illumina HiSeq4000 (Illumina, San Diego, CA) in paired end mode at Michigan State University's Research Technology Support Facility. Reads were trimmed for adapter sequences and low-quality bases using Trimmomatic[58] (v0.32) and cleaned reads were normalized and used to generate strand-specific transcriptome assemblies using Trinity (v20140717)[59] with default parameters, of which, only transcripts >200 bp were retained for subsequent analyses. The longest isoform for each transcript was deemed the representative transcript. A nucleotide BLAST database was generated based on the *N. sibirica* transcriptome and genes encoding for putative NEPS, ISY and MLPL were identified using TBLASTN with *N. cataria* and *N. mussinii* enzymes as the query. For the 7R-ISY search, *L. album* ISY was also used as a query.

To generate expression abundances, plants were grown under greenhouse conditions (day temp: 22 ± 1 °C, night temp: 20 ± 1 °C, light cycle of 14 h, RT of 50% ± 5%) and replicated tissue was collected from immature leaves, mature leaves, green stems, underground stems, and flowers. RNA was isolated and prepared as above prior to sequencing on an Illumina HiSeq4000 (Illumina, San Diego, CA) to 50 nt in single end mode at Michigan State University's Research Technology Support Facility. Replicated RNAseq reads were trimmed using Cutadapt (v3.4)[60] removing bases from each end with quality scores <30 and employing a minimum read length of 30nt. Trimmed reads were then pseudo-aligned to the generated representative transcriptome and sequences of interest to generate Transcripts per Million (TPMs) using Kallisto (v0.46.2)[61] in stranded mode employing the following two parameters; -l 200 -s 20. Reads are available in the National Center for Biotechnology Information under BioProject ID PRJNA762925.

## Additional software

Additional software employed in this study are as follows. Assay graphs were made in GraphPad Prism (8.2.0). Adobe Illustrator (15.0.0) was used to generate final figures. Nucleic acid and protein sequences were analyzed/designed in Geneious Prime (2019.1.1). Chemical structures were generated in ChemDraw Professional (17.1), unless otherwise mentioned. ESPript3.0 online tool was used to generate the alignment figure[62]. Protparam (ExPasy) online tool was used to calculate MW and extinction coefficients for determining protein concentrations.

## Reporting summary

Further information on research design is available in the Nature Research Reporting Summary linked to this article.

# Data availability

The RNA-seq data for *N. sibirica* generated in this study have been deposited in the National Center for Biotechnology Information under BioProject ID PRJNA762925. The NsNEPS2 crystal structure data generated in this study have been deposited in the Protein Data Bank (PDB) under accession code 7QUJ.

The characterized genes have been deposited in GenBank under accession codes: NsNEPSL, ON010447; NsNEPS1A, ON010448; NsNEPS1B, ON010449; NsNEPS2, ON010450; NsNEPS4A, ON010451; NsNEPS4B, ON010452; NsMLPL1, ON010453; NsMLPL2, ON010454; NsMLPL3, ON010455; NsISY, ON010456; NsP5BR, ON010457; LaISY, ON010458. Additionally, data is available from the corresponding authors upon request.

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

## Acknowledgements

We would like to thank Eva Rothe for her assistance in the growth and maintenance of the *Nepeta* plants. We would also like to thank Mohamed Omar Kamileen for isolating *trans-trans* nepetalactone from *N. sibirica* and helpful discussions on in-vitro assays, RNA extraction and general procedures. We thank Delia Serna for her assistance with GC-MS equipment and Dr. Stefan Bartram for his assistance with the CD-spectrometer equipment. This work was funded by a grant from the National Science Foundation Plant Genome Research Program (IOS-1444499), the European Research Commission (788301) and the Max Planck Society. B.H. is supported by a postdoctoral fellowship from the Alexander von Humboldt Foundation.

## Author contributions

Author contributions N.J.H.L. and S.E.O'C. designed the study and wrote the manuscript; N.J.H.L. performed experiments. B.H. performed chemical synthesis and NMR, J.C.W. and D.Z. assembled the transcriptome; J.B., L.C., and C.L. solved the crystal structure of NsNEPS2; L.Ch. assisted with cloning and assays experiments; M.K. assisted with GCMS; C.E.R.L. discovered LaISY; B.R.L. assisted with design of the study; C.R.B. provided bioinformatic and sequencing support.

## Funding

## Competing interests

The authors declare no competing interests.
