## [Peer Review File · Nature Communications]

Biocatalytic routes to stereo-divergent iridoidsREVIEWER COMMENTS

Reviewer #1 (Remarks to the Author):

This is a very interesting MS describing a method controlling iridoid stereochemistry on the basis of enzyme engineering. Using a combination of natural and engineered stereo-divergent enzymes, the biosynthesis of 7 iridoid scaffold of all 8 stereoisomers was achieved. The main experiments are clearly explained and supported by seemingly good data. The English is excellent and the supporting figures very clear. This work adds to our understanding of NsNEPS2 topology that are responsible for the stereoselective synthesis of iridoid, and this presents opportunities for production of these bioactive chemicals using metabolic engineering and synthetic biology approaches.

Major Modifications

1. Abstract should be rewritten because the current version didn't present the importance and innovation, which was not easy for readers to catch the core information.
2. The title is "A biocatalytic toolkit for the control of iridoid stereochemistry". So the authors should provide clear manipulation principles, for the convenience of constructing biosynthetic systems by other researchers.
3. In Figs 2B, 3D, 4B, and 5B, iridodial side products were detected. However, elimination of these undesirable by-product formation during enzymatic biosynthesis of target iridoids and their derivatives is important for the generality of this study.
4. In our previous work, the engineered enzymes, sometimes, couldn't be solubly expressed in *E. coli*. Were all the mutants expressed in soluble form in this study? So I suggest the authors supply the SDS-PAGE result in Supplementary information and explain the engineered protein expression in main text.
5. The authors should compare the result with recent researches. Many efforts to recapitulate nepetalactol production in microbial hosts have been reported, such as *ACS Catalysis*, 2021, 11(15): 9898-9903, *Metabolic engineering*, 2019, 55: 76-84, and *Journal of Agricultural and Food Chemistry*, 2021, 69(8): 2501-2511.

Minor Modifications

1. "Homology models were built using the Swiss-Model server" - the model quality should be included in text.
2. The chromatograms, in Figs 2B, 3D, 4B, and 5B, look somewhat 'unreal', lacking any baseline noise or distortions normally seen in this kind of data.
3. I also suggest the authors number the chemicals in Figures and text, because this will be easy for readers to understand the paper. But this depends on the authors' choice.
4. Figs. S14-S17 was the NMR evidence for the structure of nepetalactone. However, presenting raw data without any interpretation of NMR signals is inadequate to convince me the peak was indeed nepetalactone.
5. On page 11, "Combinatorial biosynthesis of nepetalactol/one isomers", "nepetalactol and nepetalactone" will be better than "nepetalactol/one".
6. In Figs 2B, 3D, 4B, and 5B, "ret time" should be rewritten as "Ret time".

Reviewer #2 (Remarks to the Author):

Hernández Lozada et. al. report synthesis of different possible 7 iridoid scaffold stereoisomers using a combination of iridoid synthase (ISY) and nepetalactol-related short chain reductases (NEPS). For this they have explored, primarily, enzymes from *Nepeta* genus. To obtained structural rationale for the oxidative-cyclization of NEPS, structure of one of the members from *N. sibirica*, NsNEPS2 has been elucidated. Undoubtedly, an extensive amount of work has been done to discover enzymes, which in combination or in standalone mode produce specific 7 iridoid scaffold stereoisomers. However, the work is based on limited structural data (currently, reported structure of NsNEPS2 and previously reported NmNEPS3 and NsNEPS2 structures) it is largely driven by the hypothesis, which

at times appears self-contradictory. I have tried to highlight some of this lacuna here.

In short, the manuscript provides a phenomenological account of production of nepetalactol stereoisomers through a combination of naturally occurring NEPs from *Nepeta* genus, without much dwelling into the scientific rationale for their observations. This, in my opinion, has reduced the scientific profundity of the work. Perhaps, addressing following concerns could make the work more generic, rather than being a mere piecemeal theory.

Major concerns:

1. Authors have set out to decouple the oxidation and cyclization properties of NEPs by mutating of the active site Tyr, which is known to disrupt the oxidative activity. However, this strategy has failed in NmNEPS1 and NsNEPS1B. These enzymes are highly homologous with NsNEPS1A, in which mutation has disrupted the activity. What are the reasons for this discrepancy? Is it due to the structural variations in the enzymes or due to the mode of substrate binding? Or both?

2. Was the substrate conformation restrained during the docking of 7S-cis-cis nepetalactol in the NmNEPS3 model? If so, the argument that polarity and size of the sidechain affect the oxidation of nepetalactol doesn't fit well, as V206A and V206I show similar yields of product. If it is not constrained then how the stereochemistry of the docked ligand was confirmed?

3. By correlating cyclization activity and sequence (and/or models) of the NEPs, it is suggested that residues at 206, 104, 108, loop 150-164 play important in determining the cyclization of different stereoisomers. Is there any synergistic effect of these regions? Why the mutational studies carried out to delineate the impact of different regions is not identical in similar clade of the enzymes chosen for the study? This opens up questions like what is the role of M206 in NmNEPS4? Why the role of these residues in NsNEPS1B was not probed? What kind of variations the model of NsNEPS1B suggests, which makes it compatible for 7S-trans-trans, 7R-trans-trans and 7R-cis-trans nepetalactone? It is expected that mutational study on residues at 206 and 198 (as per NmNEPS1 numbering) should be carried out on NsNEPS1B, as it exhibits a broad spectrum of 7 iridoid scaffold cyclization. What is the generic inference? Can there be any generic correlation between stereo-elective cyclisation and or structural and sequence determinants of the enzymes? This is a very pertinent question as the authors have set out to provide a "toolkit" for stereo-divergent enzymatic synthesis of 7 iridoid scaffold stereoisomers. I feel that this is the major lacuna of the manuscript, as the inferences are not generic and modeling & mutational studies are driven to fit the observed data.

4. In combinatorial biosynthesis of nepetalactol 8-oxogeraniol was simultaneously treated with ISY and NEPS/MLPL. What would be the products if the reactions are carried out in tandem? It has been reported (Sci. Rep. 2015 Feb 5;5:8258, I am surprised that this paper was not cited) that sequential reactions alter the ratio of products formed. It is worthwhile testing few combinatorial reactions in both, sequential and simultaneous modes.

Minor concerns:

1. Page no5: The sentence "important in determining the cyclization stereo-selectivity" needs rephrasing?? Is it cyclization and stereo-selectivity?

2. Page no 13 line no 4: "Moreover, NEPSL enzymes *N. mussinii*, *N. cataria* " should be Moreover, NEPSL enzymes of *N. mussinii*, *N. cataria*

3. It would be quite helpful for the non-expert readers, if there is a supplementary table showing the cognate substrate and products (major and minor) of the enzymes reported in the study.

4. It would be good to highlight the residues in the sequence alignment (Fig S2, on which mutational studies have been carried out. Otherwise it is very difficult to correlate the observations against the sequence variations, as the residue numbering differ from enzyme to enzyme.

Reviewer #3 (Remarks to the Author):

Hernandez Lozada and colleagues present an interesting study on the biosynthesis and enzymology of iridoids, a family of plant monoterpenoids with cat attractant and insect repellent properties. Earlier studies from the same group have shown that the early steps in iridoid biosynthesis follows oxidation of geraniol to 8-oxogeraniol, activation for cyclization via reduction by stereoselective ISY, and cyclization into the bicyclic nepetalactol by NEPS. Some NEPS also oxidize nepetalactols into nepetalactones. Given the three stereo centers in nepetalactone, there are 8 possible diastereomers with 7S-cis-trans and 7S-cis-cis the most common. The authors use a combination of X-ray crystallography, rational enzyme engineering, genome mining, combinatorial biosynthesis, and chemical synthesis to generate a toolbox that is capable of producing 7 of the 8 diastereomers. This impressive display not only allows the stereoselective production of this important terpene skeleton, but provides insight into how these enzymes control cyclization and oxidation. Particularly interesting for future studies is the discovery of natural NEPS that catalyze other stereoselective cyclizations and that there is likely another type of enzyme that is responsible for setting 7R configuration since neither ISY or NEPS from *N. sibirica* gave the 7R product. Overall, this paper is well written and clear, the figures are well organized and informative, and the data analysis and conclusions are well supported by the data. This study should be of interest to readers in the fields of natural products, enzymology, chemoenzymatic synthesis and may be of future use in agricultural, commercial, or pharmaceutical applications.

A few comments:

1. The term conformation is used throughout the manuscript when describing the stereochemical isomers. Configuration would be a better term.
2. The enzyme activity/chromatography data of NmNEPS3 clearly shows that Q206 affects oxidation activity, but the docking result in Fig. 2C doesn't appear to directly support the idea since the side chain of Q206 is not pointed toward the substrate and a distance is not given here. Understandably, this is only docking.
3. In the conclusion section, the authors state that these products are chemically difficult to synthesize. References may be useful here?
4. With the current information regarding sequence-function of NEPS and ability to decouple cyclization and oxidation, do the authors think it is currently possible to analyze other NEPS from genomes and confidently predict stereoselectivity and/or monofunctionality (decoupled) vs bifunctionality (both cyclization and oxidation)?
6. Typo: "aminoacid", with no space is found twice (pg 5 and Fig. 2 legend).

Reviewer #4 (Remarks to the Author):

This manuscript describes discovery of naturally occurring enzymes that can be used for the stereoselective synthesis of nepetalactones. The elucidation of the stereoselective construction of the iridoid skeleton, which is achieved by enzymes, is a valuable finding, because the diastereo/enantioselective synthesis of the core skeleton of iridoids are difficult in organic synthesis. After minor corrections, this manuscript might be accepted by Nature Communications.

1. Figure 5A. Please check the chemical structure of 7R-trans-cis nepetalactol/nepetalactone. It seems that the structures are the same as 7R-trans-trans nepetalactol/nepetalactone.
2. The description of stereochemistry, 'S' or 'R' should be written in italic.
Page 8, the second line from the bottom, the 7*S*.
Page 13, line 12, 7*S* or 7*R*.

Response to Reviewers

REVIEWER COMMENTS

Reviewer #1:

This is a very interesting MS describing a method controlling iridoid stereochemistry on the basis of enzyme engineering. Using a combination of natural and engineered stereo-divergent enzymes, the biosynthesis of 7 iridoid scaffold of all 8 stereoisomers was achieved. The main experiments are clearly explained and supported by seemingly good data. The English is excellent and the supporting figures very clear. This work adds to our understanding of NsNEPS2 topology that are responsible for the stereoselective synthesis of iridoid, and this presents opportunities for production of these bioactive chemicals using metabolic engineering and synthetic biology approaches.

We thank the reviewer for the positive and constructive comments.

Major Modifications

1. Abstract should be rewritten because the current version didn't present the importance and innovation, which was not easy for readers to catch the core information.

Response: We agree that the abstract needed improvement. Here is the revised version:

Thousands of natural products are derived from the fused cyclopentane-pyran molecular scaffold nepetalactol. These natural products are used in an enormous range of applications that span the agricultural and medical industries. For example, nepetalactone, the oxidized derivative of nepetalactol, is known for its cat attractant properties as well as potential as an insect repellent. Most of these naturally occurring nepetalactol-derived compounds arise from only two out of the eight possible stereoisomers, 7*S*-*cis*-*trans* and 7*R*-*cis*-*cis* nepetalactols. Here we use a combination of naturally occurring and engineered enzymes to produce seven of the eight possible nepetalactol or nepetalactone stereoisomers. These enzymes serve as a biocatalytic toolkit for production of a broader range of iridoids, providing a versatile system for the diversification of this important natural product scaffold.

2. The title is "A biocatalytic toolkit for the control of iridoid stereochemistry". So the authors should provide clear manipulation principles, for the convenience of constructing biosynthetic systems by other researchers.

Response: We used the term toolkit to refer to the collection of enzymes reported in the manuscript. We did not mean toolkit to imply that we had developed completely general design principles for NEPS engineering. We have added some sentences that we hope clarify this (see yellow highlighting in revised manuscript):

Final sentence of the abstract:

"These enzymes serve as a biocatalytic toolkit for production of a broader range of iridoids, providing a versatile system for the diversification of this important natural product scaffold. "

In the introduction:

“The NEPS provide a starting point for developing a toolkit of stereo-selective cyclases to enable this.”

In the conclusion:

“Here we report the discovery of new NEPS enzymes, both naturally occurring and engineered, that can serve as a toolkit for stereo-divergent enzymatic synthesis of 7 iridoid scaffold stereoisomers.”

We used a relatively shorter title for readability purposes. If the reviewer and editor believe that this is still misleading, we propose that we could use this longer title:

Discovery and engineering of NEPS enzymes provides a biocatalytic toolkit for the control of iridoid stereochemistry

3. In Figs 2B, 3D, 4B, and 5B, iridodial side products were detected. However, elimination of these undesirable by-product formation during enzymatic biosynthesis of target iridoids and their derivatives is important for the generality of this study.

Response: We agree with the point made by the reviewer that suppressing iridodial production is indeed a major concern when pursuing biosynthesis of these compounds. We included in all these figures a control (ISY, no NEPS) that shows that the iridodial production is much higher when the NEPS enzyme is excluded. Indeed, most of our NEPS suppress the iridodial production partially (Figure 2B) or almost completely (Figures 2A, 2D, 3D, 4B, 5B).

However, it is very important to note that with the trans-fused rings, the nepetalactol undergoes spontaneous ring opening unless it is immediately converted to the nepetalactone (see Figure 2D, NmNEPS1 154SATA vs. NmNEPS1 154SATA S198L). This is not an issue with the NEPS cyclization, but instead is due to the instability of the trans ring structure. This has been described previously (see references below, which are references 34 and 35 in the manuscript).

Liblikas, I., Santangelo, E. M., Sandell, J., Baeckström, P., Svensson, M., Jacobsson, U., & Unelius, C. R. (2005). Simplified isolation procedure and interconversion of the diastereomers of nepetalactone and nepetalactol. *Journal of Natural Products*, 68(6), 886–890. <https://doi.org/10.1021/np049647d>

Dawson, G. W., Janes, N. F., Mudd, A., Pickett, J. A., Wadhams, L. J., Slawin, M. Z., & Williams, D. J. (1989). The aphid sex pheromone. *Pure and Applied Chemistry*, 61(3), 555–558. <https://doi.org/10.1351/pac198961030555>

4. In our previous work, the engineered enzymes, sometimes, couldn't be solubly expressed in E. coli. Were all the mutants expressed in soluble form in this study? So I suggest the authors supply the SDS-PAGE result in Supplementary information and explain the engineered protein expression in main text.

Response: Every native protein and mutant described in this manuscript was expressed in soluble form. The only protein that expressed in E. coli at low levels was NEPS4, which was the major reason why we did not make mutants in the NEPS4 protein. Mutants of all other NEPS were expressed in soluble form in E. coli. One key point is that NAD⁺ stabilizes the protein at room temperature, so it is important to add the protein (which is kept cold up to this point) to the assay only after the cofactor had been added to the assay mixture. The enzymes sometimes precipitated at room temperature without the NAD⁺. This important point is now noted in the methods.

“It is important to ensure that NEPS protein stocks do not warm to room temperature unless NAD⁺ is present, otherwise, protein precipitation can occur.”

We attach a representative gel of some mutants below for the reviewer and editor to examine. If the editor feels that it is helpful, we can include this in the SI, though we would need to re-run the gel and photograph at a publication quality resolution before including them in the manuscript. Our opinion was that given the fact that the expression of these proteins was not particularly remarkable or problematic, that inclusion of a gel is not necessary.

5. The authors should compare the result with recent researches. Many efforts to recapitulate nepetalactol production in microbial hosts have been reported, such as *ACS Catalysis*, 2021, 11(15): 9898-9903, *Metabolic engineering*, 2019, 55: 76-84, and *Journal of Agricultural and Food Chemistry*, 2021, 69(8): 2501-2511.

Response: This is an excellent suggestion and these references have now been added to the introduction (references 25-29).

“These enzymes can now be used in combination with existing microbial host systems to provide a set of divergent starting points for seco-iridoid and iridoid glucoside pathways^{25–29}.”

We also state at the end of the conclusion:

“To date, 7*S*-*cis*-*trans* nepetalactol, which is also formed spontaneously in the absence of any cyclase, albeit in greatly reduced yield, has been the main focus of microbial engineering efforts^{25–29}. The enzymes reported in this study can be utilized in these metabolic engineering systems to expand the iridoid-derived natural product space and potentially generate new to nature compounds.”

Minor Modifications

1. *"Homology models were built using the Swiss-Model server" - the model quality should be included in text.*

Response: We have now added these model quality values to the methods section.

2. *The chromatograms, in Figs 2B, 3D, 4B, and 5B, look somewhat 'unreal', lacking any baseline noise or distortions normally seen in this kind of data.*

Response: We agree that the baseline looks very smooth. The reason why it appears this way as presented is twofold. First, we use a substrate concentration (0.5 mM) that results in low signal to noise ratio on our GC-MS system. In addition, we use in the assay figures a geometry that compresses the y-axis in order to provide generate figures that show more assays and thus are more readable in a space restricted manuscript form. However, as we show below using as example an assay in figure 2B, we do see the baseline distortions but they tend to be in a different order of magnitude compared to the peak signals.

Figure 2B
CrISY, no NEPS

3. I also suggest the authors number the chemicals in Figures and text, because this will be easy for readers to understand the paper. But this depends on the authors' choice.

Response: We understand the complexity of the nomenclature of these molecules, and we attempted to describe them in a way that, in our best judgement, the reader can follow them most easily. However, if the editor decides that numbering the molecules is helpful, we will change this. In case we are asked to add numbers, we suggest that this should not be done as a replacement of the full notation but in addition to it.

4. Figs. S14-S17 was the NMR evidence for the structure of nepetalactone. However, presenting raw data without any interpretation of NMR signals is inadequate to convince me the peak was indeed nepetalactone.

Response: We added labels to the NMR figures to point out which peak corresponds to each carbon. We hope that this clarifies to the reader the identity of the molecule. This is included in the Supplementary information. One representative annotated spectrum is shown below.

5. On page 11, “Combinatorial biosynthesis of nepetalactol/one isomers”, “nepetalactol and nepetalactone” will be better than “nepetalactol/one”.

Response: Thank you. We have taken the suggestion and changed it.

6. In Figs 2B, 3D, 4B, and 5B, “ret time” should be rewritten as “Ret time”.

Response: We have taken the suggestion and changed the axis titles accordingly.

Reviewer #2:

Hernández Lozada et. al. report synthesis of different possible 7 iridoid scaffold stereoisomers using a combination of iridoid synthase (ISY) and nepetalactol-related short chain reductases (NEPS). For this they have explored, primarily, enzymes from Nepeta genus. To obtained structural rationale for the oxidative-cyclization of NEPS, structure of one of the members from N. sibirica, NsNEPS2 has been elucidated. Undoubtedly, an extensive amount of work has been done to discover enzymes, which in combination or in standalone mode produce specific 7 iridoid scaffold stereoisomers. However, the work is based on limited structural data (currently, reported structure of NsNEPS2 and previously reported NmNEPS3 and NsNEPS2 structures) it is largely driven by the hypothesis, which at times appears self-contradictory. I have tried to highlight some of this lacuna here.

In short, the manuscript provides a phenomenological account of production of nepetalactol stereoisomers though a combination of naturally occurring NEPs from Nepeta genus, without much dwelling into the scientific rationale for their observations. This, in my opinion, has reduced the scientific profundity of the work. Perhaps, addressing following concerns could make the work more generic, rather than being a mere piecemeal theory.

We greatly appreciate the detailed and thoughtful comments, which we address in point by point fashion below. As a general comment, the impact of all of the mutations that we made in this family of proteins are highly dependent on the context of the entire protein. While we can mechanistically rationalize the changes in activity that result for many of these mutants, the impact of specific mutations is modulated by the rest of the protein sequence. We have explicitly made this point in the conclusion of the manuscript, where we state:

“The impact of these mutations was highly dependent on the specific protein used as a starting point for engineering. Nevertheless, we could rationalize the mechanistic basis of how these mutations modulated cyclization and oxidation activities.”

1. Authors have set out to decouple the oxidation and cyclization properties of NEPs by mutating of the active site Tyr, which is know to disrupt the oxidative activity. However, this strategy has failed in NmNEPS1 and NsNEPS1B. These enzymes are highly homologous with NsNEPS1A, in which mutation has disrupted the activity. What are the reasons for this discrepancy? Is it due the structural variations in the enzymes or due to the mode of substrate binding? Or both?

Response:

We believe that it is a combination of subtle structural changes in the active site as well as the mode of substrate binding. However, as the reviewer points out– despite extensive efforts we still have limited structural information on the NEPS– it is not possible to pinpoint these extremely subtle changes. In NmNEPS1 and NsNEPS1B there is clearly a compensatory mechanism that allows oxidation even in the absence of this Tyrosine. It is possible, that as long as the substrate is positioned appropriately next to the cofactor, another residue– or water molecule– could easily serve as the base that would be required to facilitate this oxidation of the alcohol.

We modified the text to highlight this point, as follows:

“However, this did not prove to be a general strategy for decoupling of cyclization and oxidation activity in other NEPS, since mutation of the catalytic tyrosine residue in other NEPS enzymes yielded enzymes that were completely inactive or, surprisingly, unchanged in oxidation activity in NmNEPS1 and NsNEPS1B (Figure S3). In NmNEPS1 and NsNEPS1B there is clearly a compensatory, albeit subtle, mechanism that allows oxidation even in the absence of this tyrosine. One possibility is that as long as the substrate is positioned appropriately next to the cofactor, another residue- or water molecule- could serve as the base that would be required to facilitate this oxidation of the alcohol. However, with limited structural information on the NEPS it is not possible to pinpoint these subtle differences.”

2. Was the substrate conformation restrained during the docking of 7S-cis-cis nepetalactol in the NmNEPS3 model? If so, the argument that polarity and size of the sidechain affect the oxidation of nepetalactol doesn't fit well, as V206A and V206I show similar yields of product. If it is not constrained then how the stereochemistry of the docked ligand was confirmed?

Response: First, the substrate conformation was restrained during the docking.

As described in response to Reviewer 3, point 2, we agree that we may have overinterpreted the data. What we should have said is simply that this residue clearly impacts the shape of the binding pocket based on the mutagenesis, and this makes sense given the proximity of this residue to the binding pocket as shown in the structural model.

The text now reads: “Given the proximity of this residue to the binding site as evidenced by molecular docking (see distances shown in **Figure 2C**), this residue may impact the exact orientation in which the substrate binds, thereby determining whether the lactol group is positioned such that it can be oxidized by the NAD⁺ cofactor. Thus we can control dehydrogenase activity but not impact cyclization in the 7S-cis-cis cyclase NEPS3 through a single substitution at residue 206.”

Please also see additions to the legend of Figure 2.

3. *By correlating cyclization activity and sequence (and/or models) of the NEPs, it is suggested that residues at 206, 104, 108, loop 150-164 play important in determining the cyclization of different stereoisomers. Is there is any synergistic effect of these regions?*

Yes, there are synergistic effects taking place and several of our findings demonstrate this. Moreover, these synergistic effects do not translate to the same outcome in the same protein. We will highlight some of these effects below.

1. NEPS1 vs NEPS4 results (Figures 2D and S6): In this example, NmNEPS1 can be seen to gain cyclization of 7S-trans-cis nepetalactol when mutations 154SATA and S198L are performed. Then we show that S198L was preventing further oxidation of the 7S-trans-cis nepetalactol to nepetalactone. When S198 residue is left unchanged NmNEPS1 154SATA is able to perform both, cyclization and oxidation of 7S-trans-cis nepetalactone.
2. But the role of S198 is not solely to allow or disallow oxidation of 7S-trans-cis nepetalactol. In Figure S8 we show that the “wrong” residue 198 can also have deleterious effects on the ability to cyclize 7S-trans-cis nepetalactol (154SATA S198P and 154SATA S198M, for example).
3. NmNEPS4 loop swap results in different outcomes when it is placed in NmNEPS1 (Figure 2D) vs NcNEPS3A (Figure S5). In NmNEPS1 this replacement results in some 7S-trans-cis nepetalactone being produced, whereas in the context of NcNEPS3A it results in some 7S-cis-cis nepetalactone being made. The loop is able to perform cyclization but the product varies depending on the context of the enzyme.
4. NmNEPS1 154SATA loop variations (Figure S7). It is shown here, even when the polar residues S154 and T156 are left intact, slight changes in the context around them (A153 or A155 changes to G, P, V or S) have influence on the production of 7S-trans-cis nepetalactone.
5. We show the importance of residue V206 in NcNEPS3A for its ability to oxidize 7S-cis-cis nepetalactol to nepetalactone. However, this residue does not play a role in NmNEPS1 and NmNEPS4 since it is conserved as a Methionine.

Other examples of this can be seen through the manuscript, highlighting that no NEPS is equal to the others, and mutations should serve as a guide for exploration but not as a “recipe” for predicting specific activities. While we absolutely agree with the reviewer that general design principles for these proteins would be highly desirable, it is clear that the effect of specific mutations is dependent on the context of the entire protein.

Why the mutational studies carried out to delineate the impact of different regions is not identical in similar clade of the enzymes chosen for the study? This opens up questions like what is the role of M206 in NmNEPS4?

In this manuscript, we did not attempt to turn every NEPS into every other NEPS. Here, we successfully switched NmNEPS1 and NmNEPS4 activity (in other words, we made a NEPS1 mutant with trans-cis cyclization). Since this methionine is conserved between NmNEPS1 and NmNEPS4, mutating this residue did not make sense in this context.

Why the role of these residues in NsNEPS1B was not probed?

The only mutant that wasn't tested in NEPS1B was Q206, because this residue is highly conserved across the NEPS, and therefore, outside of NEPS3, it would not be playing any role in mediating product

selectivity. Please see supplementary figure S9 for the traces of all NsNEPS1B mutants and Figure S2 for the alignment.

What kind of variations the model of NsNEPS1B suggests, which makes it compatible for 7S-trans-trans, 7R-trans-trans and 7R-cis-trans nepetalactone?

We really appreciate this question: if we could answer this clearly, we would be able to have a clear-cut set of rules for engineering! We were able to make mutations to key residues that we know are involved in cyclization that rendered NsNEPS1B incapable of cyclizing to 7S-trans-trans and 7R-trans-trans. However, as it is common with biocatalysts, we found that the background protein being mutagenized, even when it is relatively high in protein sequence could yield different catalytic behavior. What we hope to convince the reader is that the residues we propose as important indeed have a strong impact in this behavior.

It is expected that mutational study on residues at 206 and 198 (as per NmNEPS1 numbering) should be carried out on NsNEPS1B, as it exhibits a broad spectrum of 7 iridoid scaffold cyclization. What is the generic inference?

We did perform the 198 mutation (see Figure S9, S195L). For residue 206, please see comments above.

Can there be any generic correlation between stereo-elective cyclisation and or structural and sequence determinants of the enzymes? This is a very pertinent question as the authors have set out to provide a “toolkit” for stereo-divergent enzymatic synthesis of 7 iridoid scaffold stereoisomers. I feel that this is the major lacuna of the manuscript, as the inferences are not generic and modeling & mutational studies are driven to fit the observed data.

See response to question 4 of reviewer 3

4. In combinatorial biosynthesis of nepetalactol 8-oxogeranial was simultaneously treated with ISY and NEPS/MLPL. What would be the products if the reactions are carried out in tandem? It has been reported (Sci. Rep. 2015 Feb 5;5:8258, I am surprised that this paper was not cited) that sequential reactions alter the ratio of products formed. It is worthwhile testing few combinatorial reactions in both, sequential and simultaneous modes.

Response: First, we agree that we should cite the Scientific Reports paper and it is now included as reference 17.

However, the final product options for tandem reactions is very limited since NEPS cyclization requires an unstable intermediate. In the absence of NEPS, ISY generates an unstable intermediate, 8-oxocitronellyl enol, that is spontaneously converted towards various iridodials and cis-trans-nepetalactol. If NEPS were to be added after this takes place, the only additional possibility is that oxidation of cis-trans-nepetalactol to cis-trans-nepetalactone could occur. NEPS cannot catalyze additional cyclization (for example conversion of iridodials to nepetalactols) or conversion of nepetalactols or iridodials to other stereoisomers. This was shown in Supplementary Figure 11 of Lichman et al. (see below).

Lichman, A. B. R., Kamileen, M. O., Titchiner, G. R., Saalbach, G., Clare, E., Stevenson, M., ... O'Connor, S. E. (2018). Uncoupled activation and cyclisation in catmint reductive terpenoid biosynthesis. *Nat Chem Biol*, 5(January), 1–16. <https://doi.org/10.1101/391953>

Minor concerns:

1. Page no5: The sentence “important in determining the cyclization stereo-selectivity” needs rephrasing?? Is it cyclization and stereo-selectivity?

Response: Thank you for the suggestion. We are trying to convey that the mutations made by replacing the loop from NmNEPS3A to that of NEPS4 did have an impact in the product profile of NsNEPS3A, they were not sufficient to generate a NEPS4-type variant.

We rephrased the sentence to: “We concluded that while the 150-162 loop is important in determining stereo-selectivity, it does not fully control it. Therefore, we needed to examine the role other regions of the binding pocket play in determining specificity.”

2. Page no 13 line no 4: “Moreover, NEPSL enzymes *N. mussinii*, *N. cataria* “should be Moreover, NEPSL enzymes of *N. mussinii*, *N. cataria*

Response: We have corrected this.

3. It would be quite helpful for the non-expert readers, if there is a supplementary table showing the cognate substrate and products (major and minor) of the enzymes reported in the study.

Response: This is an excellent suggestion. See new Supplementary Tables 4 and 5.

4. It would be good to highlight the residues in the sequence alignment (Fig S2, on which mutational studies have been carried out. Otherwise it is very difficult to correlate the observations against the sequence variations, as the residue numbering differ from enzyme to enzyme.

Response: Thank you for the suggestion. We have modified the figure to include additional information that will allow the reader to track where the changes were made.

Reviewer #3:

Hernandez Lozada and colleagues present an interesting study on the biosynthesis and enzymology of iridoids, a family of plant monoterpenoids with cat attractant and insect repellent properties. Earlier studies from the same group have shown that the early steps in iridoid biosynthesis follows oxidation of geraniol to 8-oxogeraniol, activation for cyclization via reduction by stereoselective ISY, and cyclization into the bicyclic nepetalactol by NEPS. Some NEPS also oxidize nepetalactols into nepetalactones. Given the three stereo centers in nepetalactone, there are 8 possible diastereomers with 7S-cis-trans and 7S-cis-cis the most common. The authors use a combination of X-ray crystallography, rational enzyme engineering, genome mining, combinatorial biosynthesis, and chemical synthesis to generate a toolbox that is capable of producing 7 of the 8 diastereomers. This impressive display not only allows the stereoselective production of this important terpene skeleton, but provides insight into how these enzymes control cyclization and oxidation. Particularly interesting for future studies is the discovery of natural NEPS that catalyze other stereoselective cyclizations and that there is likely another type of enzyme that is responsible for setting 7R configuration since neither ISY or

NEPS from N. sibirica gave the 7R product. Overall, this paper is well written and clear, the figures are well organized and informative, and the data analysis and conclusions are well supported by the data. This study should be of interest to readers in the fields of natural products, enzymology, chemoenzymatic synthesis and may be of future use in agricultural, commercial, or pharmaceutical applications.

A few comments:

1. The term conformation is used throughout the manuscript when describing the stereochemical isomers. Configuration would be a better term.

Response: Thank you for the suggestion. We changed the use of the term “conformation” to “configuration”. Changes are highlighted in yellow.

2. The enzyme activity/chromatography data of NmNEPS3 clearly shows that Q206 affects oxidation activity, but the docking result in Fig. 2C doesn't appear to directly support the idea since the side chain of Q206 is not pointed toward the substrate and a distance is not given here. Understandably, this is only docking.

Response: We agree with the reviewer that the side chain of Q206, while close to the binding pocket, is not pointing directly at the substrate in these molecular docking experiments. Nevertheless, as the reviewer points out, the biochemical data clearly shows that this residue controls oxidation. We have revised the text to simply say that Q/V206 is part of the substrate binding pocket, and therefore, mutagenesis of this residue likely impacts the binding orientation of the substrate, which would in turn impact whether or not it can be oxidized by the NAD⁺ cofactor.

The text now reads:

“Given the proximity of this residue to the binding site as evidenced by molecular docking (see distances shown in **Figure 2C**), this residue may impact the exact orientation in which the substrate binds, thereby determining whether the lactol group is positioned such that it can be oxidized by the NAD⁺ cofactor. Thus we can control dehydrogenase activity but not impact cyclization in the 7*S*-*cis-cis* cyclase NEPS3 through a single substitution at residue 206.”

See also response to Reviewer 2, point 2.

3. In the conclusion section, the authors state that these products are chemically difficult to synthesize. References may be useful here?

Response: We have added references that highlight the difficulty of synthesizing these molecules in stereospecific manner (references 38-40).

4. With the current information regarding sequence-function of NEPS and ability to decouple cyclization and oxidation, do the authors think it is currently possible to analyze other NEPS from genomes and confidently predict stereoselectivity and/or monofunctionality (decoupled) vs bifunctionality (both cyclization and oxidation)?

Response: We appreciate what the reviewer is asking. Using the Q206V mutation as an example, you can see that this residue switches from Q to V in NmNEPS3 to NcNEPS3A (as shown in Figure S2). But in

all of the other NEPS, residue 206 is a methionine. Therefore, the switch from Q to V only switches the oxidation activity in the context of the NEPS3 protein background. This is true for many of the mutations that we made in this family of proteins; the effect of the mutations are highly dependent on the context of the entire protein. We recognize that it is satisfying to have clear cut design rules, but this simply was not possible here, though we hope that we have provided rational mechanistic explanations regarding the changed activity of our reported mutations.

We have explicitly made this point in the conclusion of the manuscript where we now state:

“The impact of these mutations was highly dependent on the specific protein used as a starting point for engineering. Nevertheless, we could rationalize the mechanistic basis of how these mutations modulated cyclization and oxidation activities.”

6. Typo: “aminoacid”, with no space is found twice (pg 5 and Fig. 2 legend).

Response: We have corrected these typos.

Reviewer #4:

This manuscript describes discovery of naturally occurring enzymes that can be used for the stereoselective synthesis of nepetalactones. The elucidation of the stereoselective construction of the iridoid skeleton, which is achieved by enzymes, is a valuable finding, because the diastereo/enantioselective synthesis of the core skeleton of iridoids are difficult in organic synthesis. After minor corrections, this manuscript might be accepted by Nature Communications.

1. Figure 5A. Please check the chemical structure of 7R-trans-cis nepetalactol/nepetalactone. It seems that the structures are the same as 7R-trans-trans nepetalactol/nepetalactone.

Response: Thank you for noticing this. It is now fixed.

2. The description of stereochemistry, ‘S’ or ‘R’ should be written in italic.

Page 8, the second line from the bottom, the 7*S*.

Page 13, line 12, 7*S* or 7*R*.

Response: Thank you for noticing this. We fixed both of the specific examples pointed out by the reviewer and revised/fixed any additional instance of this in the manuscript and supplementary info.

REVIEWER COMMENTS

Reviewer #1 (Remarks to the Author):

This paper described a “toolkit” for stereo-divergent enzymatic synthesis of 7 iridoid scaffold stereoisomers. But, the MS didn't present a general method, which could be utilized by other researchers. Please summarize and refine the generality of this toolkit.

Reviewer #2 (Remarks to the Author):

In the revised manuscript Hernández Lozada et. al., have tried to address some of the concerns raised by this reviewer. Although, few concerns have been addressed satisfactorily, the major ones have been dealt superficially.

For example, when the manuscript falls short in explaining the generic design principles for engineering the enzymes then the word “tool kit” stands out as a misnomer. As response to the concerns raised by this, as well as, by the other reviewers, they state that the word “toolkit” is meant to describe the set of enzymes for the production of a broader range of iridoids. This doesn't justify the usage of the word “toolkit” as the working principles of these “kits” are not known.

Furthermore, in the conclusion section, it has been stated, “Nevertheless, we could rationalize the mechanistic basis of how these mutations modulated cyclization and oxidation activities”.

Given the limited structural information, mechanistic explanations provided for the enzymes are more of hypothesis than rationale.

In short, lot of data presented here is over interpreted with eulogizing terminologies, such as “toolkit”. This reviewer is of the opinion that the word “toolkit” must be dropped from the manuscript.

Reviewer #3 (Remarks to the Author):

The authors addressed all of my questions/concerns

REVIEWER COMMENTS

Reviewer #1 and Reviewer #2

This paper described a “toolkit” for stereo-divergent enzymatic synthesis of 7 iridoid scaffold stereoisomers. But, the MS didn't present a general method, which could be utilized by other researchers. Please summarize and refine the generality of this toolkit.

In the revised manuscript Hernández Lozada et. al., have tried to address some of the concerns raised by this reviewer. Although, few concerns have been addressed satisfactorily, the major ones have been dealt superficially. For example, when the manuscript falls short in explaining the generic design principles for engineering the enzymes then the word “tool kit” stands out as a misnomer. As response to the concerns raised by this, as well as, by the other reviewers, they state that the word “toolkit” is meant to describe the set of enzymes for the production of a broader range of iridoids. This doesn't justify the usage of the word “toolkit” as the working principles of these “kits” are not known. Furthermore, in the conclusion section, it has been stated, “Nevertheless, we could rationalize the mechanistic basis of how these mutations modulated cyclization and oxidation activities”. Given the limited structural information, mechanistic explanations provided for the enzymes are more of hypothesis than rationale. In short, lot of data presented here is over interpreted with eulogizing terminologies, such as “toolkit”. This reviewer is of the opinion that the word “toolkit” must be dropped from the manuscript.

It was not our intention to be misleading with the use of the word toolkit, and we apologize for this word; our interpretation of this word did not match with other readers. We have stripped the manuscript of this word, and we have also changed the title. The engineering of the NEPS is not generalizable; we can rationalize the effect of certain mutations, but these mutations do not always exert the same effects in the context of different NEPS. However, we sincerely hope that the biocatalysis and natural products community will be highly interested in these proteins that have these different combinations of stereoselectivity and oxidation capabilities.

To fully address the concerns of Reviewer 1 and 2, we have, as mentioned above, removed all mentions of the word toolkit. We have also, in the conclusion, replaced “rationale” with “hypothesis” which we agree with Reviewer 2 is a better word in this context. Regarding Reviewer 1's request for a summary paragraph, please see the highlighted sentences in the conclusion. The “generality” of the method for engineering is limited, and we now state this explicitly. I think that our new text addresses Reviewer 1's request; we did not think that it would be appropriate to try to recapitulate all of the mutation effects in the final concluding sections, which we thought should be a relatively short section. Please see the excerpted revised text in the conclusion below:

The discovery of a group of naturally occurring orthologues with varying product selectivity served as an excellent starting point for modulating oxidative and cyclization selectivity. The impact of the engineered mutations was often dependent on the specific protein. While this precluded formulating a generalizable set of design rules for engineering these enzymes, we nevertheless could make some hypotheses

regarding the mechanistic basis of how these mutations modulated cyclization and oxidation activities.
Finally, we could productively combine cyclases to lead to a non-naturally occurring stereoisomer.

Reviewer #3 (Remarks to the Author):

The authors addressed all of my questions/concerns

REVIEWERS' COMMENTS

Reviewer #1 (Remarks to the Author):

I agree to the publication of this manuscript in Nature Communications.

Reviewer #2 (Remarks to the Author):

Authors have addressed my concerns in the revised manuscript, satisfactorily.